# Efficient Network Automatic Relevance Determination

**Hongwei Zhang** [1 2 3] **Ziqi Ye** [4 3] **Xinyuan Wang** [5 3] **Xin Guo** [3] **Zenglin Xu** [1 3] **Yuan Cheng** [1 3] **Zixin Hu** [1 3]
**Yuan Qi** [1 6 3]

## Abstract

We propose Network Automatic Relevance Determination (NARD), an extension of ARD for linearly probabilistic models, to simultaneously model sparse relationships between inputs $X \in \mathbb{R}^{d \times N}$ and outputs $Y \in \mathbb{R}^{m \times N}$, while capturing the correlation structure among the $Y$. NARD employs a matrix normal prior which contains a sparsity-inducing parameter to identify and discard irrelevant features, thereby promoting sparsity in the model. Algorithmically, it iteratively updates both the precision matrix and the relationship between $Y$ and the refined inputs. To mitigate the computational inefficiencies of the $\mathcal{O}(m^3 + d^3)$ cost per iteration, we introduce Sequential NARD, which evaluates features sequentially, and a Surrogate Function Method, leveraging an efficient approximation of the marginal likelihood and simplifying the calculation of determinant and inverse of an intermediate matrix. Combining the Sequential update with the Surrogate Function method further reduces computational costs. The computational complexity per iteration for these three methods is reduced to $\mathcal{O}(m^3 + p^3)$, $\mathcal{O}(m^3 + d^2)$, $\mathcal{O}(m^3 + p^2)$, respectively, where $p \ll d$ is the final number of features in the model. Our methods demonstrate significant improvements in computational efficiency with comparable performance on both synthetic and real-world datasets.

---

[1]Artificial Intelligence Innovation and Incubation Institute, Fudan University, Shanghai, China [2]School of Data Science, Fudan University, Shanghai, China [3]Shanghai Academy of Artificial Intelligence for Science, Shanghai, China [4]Department of Applied Mathematics and Theoretical Physics, University of Cambridge, Cambridge, United Kingdom [5]Eberly College of Science, Pennsylvania State University, PA, United State [6]Zhongshan Hospital, Fudan University, Shanghai, China. Correspondence to: Zixin Hu <huzixin@fudan.edu.cn>, Yuan Qi <qiyuan@fudan.edu.cn>.

*Proceedings of the 42$^{nd}$ International Conference on Machine Learning*, Vancouver, Canada. PMLR 267, 2025. Copyright 2025 by the author(s).

## 1. INTRODUCTION

Multiple-input Multiple-output Regression is a powerful modeling framework widely applied in quantitative disciplines such as finance and genomics (Zellner, 1962; Breiman & Friedman, 1997; Hastie et al., 2009; Wang, 2010; He et al., 2016). In biological research, this approach is often used to explore how molecular-level features (micro-phenotypes) influence broader phenotypic traits (macro-phenotypes) (Stephens, 2013; Akbani et al., 2014). A typical example involves examining how gene expression levels or protein concentrations, influence disease states or developmental outcomes. However, biological data often involve ultra-high-dimensional features, with thousands of genes or proteins contributing to a limited number of observable samples (Moon et al., 2019). Such high-dimensional settings present significant computational challenges.

Empirical studies suggest that only a subset of specific molecular features significantly impacts the observed phenotypes. This highlights the need for sparse regression models that focus on the most relevant features, reducing over-fitting and enhancing interpretability. Furthermore, macro-phenotypes often exhibit sparse network structures, where only a few phenotypes are connected via interaction relationship. This sparse structure highlights the need for approaches that capture not only the individual effects of inputs but also the inter-dependencies among multiple outputs. Emerging methodologies for multi-omics integration and cross-modal network information processing capitalize on this principle, reflecting the fact that biological mechanisms arise from the complex interplay of numerous molecular events and their interactions (Cohen et al., 2022; Kristensen et al., 2014).

Another related example is the study of expression quantitative trait loci (eQTL). Some genetic variants can affect the expression of multiple genes, acting as potential confounders in gene networks. Gene expression data alone are unable to fully capture the gene activities. Ignoring these effects can result in spurious associations, leading to false positives or false negatives. Incorporating covariates such as genetic variants from eQTL studies improves the estimation of relationships among genes at the transcriptional level.

In this paper, we focus on linearly probabilistic models

(Minka, 2000). Drawing inspiration from *Automatic Relevance Determination* (ARD) framework (MacKay, 1992; MacKay et al., 1994), we introduce an extension named *Network ARD* (NARD) for multiple output regression. Specifically, we place an ARD prior on the regression coefficient matrix, enabling us to determine which input features are relevant for predicting the outputs. Simultaneously, we apply an $L_1$ penalty on the precision matrix to encourage sparsity, thereby modeling the dependencies among the output. Although ARD prior is effective for feature selection, it faces computational challenges in high-dimensional settings. Standard ARD methods optimized via Expectation-Maximization (EM) or type-II maximum likelihood incur an $\mathcal{O}(d^3)$ computational cost due to matrix inversion, where $d$ is the number of features.

To address this issue, we design several novel algorithms within the NARD framework. Specifically, inspired by Tipping's greedy approach (Tipping & Faul, 2003), we develop the sequential update method, which sequentially adds and removes features. This approach allows the model to start with a few features, enabling decisions about each new feature based on its contribution to the overall evidence. Additionally, we introduce a surrogate function that approximates the lower bound of the log marginal likelihood, avoiding the need for matrix inversion. By integrating these two approaches, we provide a more efficient implementation of NARD, making it scalable for high-dimensional datasets.

In summary, we present the NARD framework, which jointly estimates the sparse regression coefficients and the precision matrix. To improve computational efficiency, we propose three novel extensions: Sequential NARD, Surrogate NARD and Hybrid NARD, which reduce complexity to $\mathcal{O}(m^3 + p^3)$, $\mathcal{O}(m^3 + d^2)$ and $\mathcal{O}(m^3 + p^2)$, respectively. These approaches achieve significant improvements in computational efficiency while maintaining comparable predictive performance in synthetic and real-world datasets.

### 1.1. Problem formulation

Under the framework of linearly probabilistic models, given an input $x \in \mathbb{R}^d$ and an output $y \in \mathbb{R}^m$, we consider

$$y = Wx + \epsilon, \tag{1}$$

where $W \in \mathbb{R}^{m \times d}$ is the regression coefficient matrix and $\epsilon \sim \mathcal{N}(0, V)$ represents the error term, assumed to follow a normal distribution with mean zero and covariance matrix $V$. Given $N$ sample pairs, the model extends to:

$$Y_{m \times N} = W_{m \times d} X_{d \times N} + \mathcal{E}_{m \times N}, \tag{2}$$

where the $i$-th column of $\mathcal{E}$ is $\epsilon$.

Our goal is to jointly estimate the regression coefficient $W$ and the precision matrix $\Omega = V^{-1}$. A typical approach is the maximum likelihood estimator (MLE). The negative log-likelihood function of $Y$, up to a constant, is given by

$$l(W, \Omega) = \mathbf{Tr}[(Y - WX)^\top (Y - WX)\Omega] - N \log |\Omega|. \tag{3}$$

It is noticed that $l(W, \Omega)$ is not jointly convex in $W$ and $\Omega$, but is bi-convex, i.e., it is convex in $W$ for fixed $\Omega$ and in $\Omega$ for fixed $W$. A common approach to address this bi-convexity is to employ an alternating minimization strategy, also known as the block coordinate descent (BCD) method (Tseng, 2001). This technique iteratively updates the parameters by fixing one set while optimizing the other until convergence is reached.

### 1.2. Related work

**Joint mean–covariance estimation.** The problem of joint multivariate variable and covariance selection has garnered significant attention in recent years. Rothman et al. (2010) proposed MRCE, which incorporates an $l_1$ penalty to promote sparsity in both $W$ and $\Omega$. This formulation leads to a BCD algorithm for solving the problem. Formally, the penalized log-likelihood is

$$\left(\hat{W}, \hat{\Omega}\right) = \underset{(W, \Omega)}{\operatorname{argmin}} \left\{ l(W, \Omega) + \lambda_1 \sum_{k \neq \ell} |\omega_{k\ell}| + \lambda_2 \sum_{j=1}^{md} |w_j| \right\}, \tag{4}$$

where $\omega_{k\ell}$ are the elements of $\Omega$, $w_j$ are the elements of vectorized $W$, $\lambda_1, \lambda_2 > 0$ are tuning parameters. This formulation yields an alternative lasso (Tibshirani, 1996) and a graphical lasso problem. Similar approaches have been explored in previous studies, such as (Cai et al., 2013; Chen et al., 2016; Lin et al., 2016; Zhang & Schneider, 2010; Zhao et al., 2020). Another approach is to adopt the Bayesian framework, where a hierarchical model is constructed using appropriate priors, followed by MCMC sampling for parameter updates and probabilistic uncertainty quantification, as seen in (Bhadra & Mallick, 2013; Deshpande et al., 2019; Li et al., 2021; Ha et al., 2021; Samanta et al., 2022). The Bayesian methods provide a principled way to handle model uncertainty while promoting sparsity. Compared to penalized methods, MCMC-based techniques tend to be computationally expensive, making them less efficient for large-scale problems.

**ARD and SBL.** Automatic Relevance Determination (ARD), closely related to Sparse Bayesian Learning (SBL) (Tipping, 2001; Faul & Tipping, 2001; Wipf & Rao, 2004), is a framework designed to identify and discard irrelevant features from high-dimensional data. ARD leverages a parameterized, data-driven prior to promote sparsity, mitigating the ill-posed nature of problems.

Subsequent studies have theoretically extended ARD or SBL by establishing connections with iterative reweighted $l_1$ minimization (Wipf & Nagarajan, 2007; 2010), compressive

sensing (Babacan et al., 2009), stepwise regression (Ament & Gomes, 2021), or by developing new efficient iterative algorithms (Al-Shoukairi et al., 2017; Zhou et al., 2021; Wang et al., 2024).

**Graphical Lasso.** The Graphical Lasso (GLasso) aims to estimate a sparse precision matrix $\Theta$ for a given dataset (Friedman et al., 2008; Banerjee et al., 2008; Yuan & Lin, 2007). The general form of GLasso is defined as:

$$\hat{\Omega} = \arg\min_{\Omega} \left( -\log|\Omega| + \mathbf{Tr}(\tilde{V}\Omega) + \psi_\lambda(\Omega) \right), \quad (5)$$

where $\hat{\Omega}$ is the estimated precision matrix, $\tilde{V}$ is the empirical covariance matrix and $\psi_\lambda(\Omega)$ is the regularization term controlling the sparsity of the solution with strength parameter $\lambda$. When $\psi_\lambda(\Omega) = \lambda \sum_{i \neq j} |\omega_{ij}|$, Eq. (5) represents the original GLasso (Friedman et al., 2008). Other representative choices for $\psi_\lambda(\Omega)$ include the adaptive LASSO or SCAD penalty (Fan et al., 2009) and graphical horseshoe (Li et al., 2019) in the Bayesian framework.

# 2. NETWORK AUTOMATIC RELEVANCE DETERMINATION

## 2.1. Framework of NARD

In this paper, we impose ARD prior on the regression coefficient matrix and $L_1$ penalty on the precision matrix to encourage sparsity. Actually, there are other potential penalties that can be applied to the precision matrix, offering flexibility in model specification. These alternatives can be considered as plug-in options.

Specifically, we impose the matrix normal distribution as the prior of $W$. The probability density function is given by

$$W \sim \mathcal{MN}(0, V_{m \times m}, K_{d \times d}^{-1})$$
$$= \frac{|K|^{m/2}}{(2\pi)^{md/2}|V|^{d/2}} \times \exp\left[ -\frac{1}{2}\mathbf{Tr}(V^{-1}WKW^\top) \right]. \quad (6)$$

where $V$ and $K^{-1}$ are two positive definite matrices representing the covariance matrices for rows and columns of $W$ respectively. In this paper, we restrict $K = \text{diag}(\alpha_1, \cdots, \alpha_d)$ to be a diagonal matrix.

The basic idea of ARD is to give the $W_{ij}$ independent parameterized data-dependent priors. The hyperparameters $V$ and $K$ are trained from the data by maximizing the evidence $P(Y|X, V, K)$, which can be done via type-II maximum likelihood or Expectation–Maximization (EM) algorithm.

$$p(Y, V, K|X) = \int p(Y|W, X)\, p(W|V, K)\, p(V)p(K)dw. \quad (7)$$

Here $p(V)$ and $p(K)$ are hyperpriors imposed on $V$ and $K$, which will be specified later. To estimate $V, K, W$, an essential procedure is to maximize the marginal likelihood

function (MLF) in Eq. (7). This is equivalent to minimize the negative logarithm of the MLF.

Recall $l(W, \Omega)$ is not jointly convex in $W$ and $\Omega$, we employ the BCD method to iteratively update the parameters. This involves alternatively solving the sparse covariance estimation problem and performing Bayesian linear regression with the ARD prior, cycling through the parameters until convergence is reached.

## 2.2. Evidence approximation

Referring to the notation in (Minka, 2000), we define

$$\begin{aligned}
S_{xx} &= XX^\top + K, \\
S_{yy} &= YY^\top, \\
S_{yx} &= YX^\top, \\
S_{y|x} &= S_{yy} - S_{yx}S_{xx}^{-1}S_{yx}^\top.
\end{aligned} \quad (8)$$

By multiplying the likelihood function $p(Y|W, X)$ with the conjugate prior Eq. (6) of $W$, i.e., $p(W|V, K)$, we have:

$$\begin{aligned}
&\ln p(Y, W|X, V, K) \\
&\propto \mathbf{Tr}\left[ V^{-1}\left(WS_{xx}W^\top - 2S_{yx}W^\top + S_{yy}\right) \right] \\
&= \mathbf{Tr}\Big[ V^{-1}\left(W - S_{yx}S_{xx}^{-1}\right)S_{xx}\left(W - S_{yx}S_{xx}^{-1}\right)^\top \\
&\quad + V^{-1}S_{y|x} \Big].
\end{aligned} \quad (9)$$

As a result, the posterior distribution for the coefficient matrix $W$ remains a matrix normal distribution and the MLF for $Y$ can then be given by integrating out $W$ from the joint distribution given in Eq. (9). Formally, we have:

$$\begin{aligned}
p(W|X, Y, V, K) &= \mathcal{MN}(\mu, V, \Sigma), \\
p(Y|X, V, K) &= \mathcal{MN}(0, V, C),
\end{aligned} \quad (10)$$

where $\mu = S_{yx}S_{xx}^{-1}$ and $\Sigma = S_{xx}$ are the posterior mean and column covariance of $W$, respectively. $C = I + X^\top K^{-1}X$ is the columns covariance of $Y$.

The negative logarithm of the MLF takes the form:

$$\begin{aligned}
\mathcal{L} &= -\ln p(Y|X, V, K) \\
&\propto m \ln|C| + N\ln|V| + \mathbf{Tr}(Y^\top V^{-1}YC^{-1}).
\end{aligned} \quad (11)$$

In practice we use Woodbury identity to calculate $C^{-1}$:

$$C^{-1} = (I + X^\top K^{-1}X)^{-1} = I - X^\top S_{xx}^{-1}X. \quad (12)$$

Note that the term $YC^{-1}Y^\top$ in Eq. (11) can be re-expressed by introducing the latent variable $W$ as follows:

$$\begin{aligned}
YC^{-1}Y^\top &= (Y - \mu X)(Y - \mu X)^\top + \mu K\mu^\top \\
&= \min_W (Y - WX)(Y - WX)^\top + WKW^\top,
\end{aligned} \quad (13)$$

i.e., $\ln p(Y|X,V,K) = \ln p(Y|X,V,K,W)$.

Then we use graphical lasso to update $V$, i.e.,

$$\hat{V}, \hat{V}^{-1} \leftarrow \textbf{glasso}(V, \lambda), \qquad (14)$$

where **glasso** refers to GLasso procedure, which takes the empirical covariance as input and returns updated covariance $\hat{V}$ and precision matrix $\hat{V}^{-1}$. The parameter $\lambda$ controls the sparsity of the $\hat{V}^{-1}$ via the $L_1$ penalty.

### 2.3. Algorithm summary

We can incorporate hyperpriors for the parameters $V$ and $K$ in our model. The hierarchical Bayesian approach allows for greater flexibility and robustness by capturing uncertainties in the prior distributions. We have $p(W; V) = \int p(W|K; V)p(K)d\alpha$. If each $\alpha_i$ follows a common Gamma distribution, then the prior of $W_i$ is a multidimensional Student distribution.

When a specific prior is adopted, $\alpha$ and $V$ reach their minima at the points where the gradient of the logarithm of the MLF is zero. It can be shown that

$$V^{\text{new}} = \frac{(Y - \mu X)(Y - \mu X)^\top + \mu K \mu^\top}{N}. \qquad (15)$$

Here we consider two different priors for $\alpha$.

**Flat prior.** If $\alpha$ follows the flat prior, i.e., $p(\alpha) = 1$ then

$$\alpha_i^{\text{new}} = \frac{m}{m(S_{xx}^{-1})_{ii} + (\mu^\top V^{-1} \mu)_{ii}}. \qquad (16)$$

**Gamma prior.** If $\alpha$ follows a Gamma prior $\text{Gamma}(a, b)$:

$$p(\alpha) = \prod_{i=1}^{d} \frac{b^a}{\Gamma(a)} \alpha_i^{a-1} e^{-b\alpha_i}. \qquad (17)$$

where $a$ and $b$ are the shape and rate parameters of the Gamma distribution, respectively. Then

$$\alpha_i^{\text{new}} = \frac{m + 2a - 2}{m(S_{xx}^{-1})_{ii} + (\mu^\top V^{-1} \mu)_{ii} + 2b}. \qquad (18)$$

Note that if any $\alpha_i = 0$, then $W_{\cdot i} = 0$ and the corresponding feature is effectively pruned from the model. In each iteration, we calculate $\Sigma = S_{xx}^{-1} = (K + XX^\top)^{-1}$, $\mu = YX^\top\Sigma$ and update $\alpha$, $V$ as the above formula. We repeat the process above until the number of iterations reaches the maximum or $\max|\Delta(\frac{1}{\alpha}, t)| := \max|\frac{1}{\alpha^{(t)}} - \frac{1}{\alpha^{(t-1)}}|$ is significantly small, where $t$ is the iteration number. Algorithm 1 summarizes the proposed NARD.

**Prior for $W$.** A conjugate prior for $V$ is the inverse Wishart distribution. This will lead to a matrix-$\mathcal{T}$ distribution (Gupta & Nagar, 2018) for the MLF of $Y$. Although this will not

---

**Algorithm 1** NARD

**Input:** Input data $X, Y, \epsilon$
**Output:** Estimated $\alpha, V, V^{-1}, W$
1: Initialize $\alpha$ elements, $V \leftarrow \hat{V}_{\text{MLE}}$.
2: **for** $t \leftarrow 1$ **to** $T$ **do**
3:     Compute $\Sigma \leftarrow (K + XX^\top)^{-1}$.
4:     Compute $\mu \leftarrow YX^\top\Sigma$.
5:     Update $V$ according to (15).
6:     $V, V^{-1} \leftarrow \textbf{glasso}(V, \lambda)$.
7:     Update $\alpha$ according to (16), (18) depending on whether the Gamma prior is included.
8:     **if** $\max|\Delta(\frac{1}{\alpha})| \le \epsilon$ **then**
9:         **Break**.
10:     **end if**
11:     Update $W \leftarrow \mu$.
12: **end for**

---

affect the main conclusions and derivations of the paper, it may complicate the expressions for some variables. Due to space limitations, we focus solely on the prior for $K$. However, the technical conclusions of this paper remain applicable to the prior for $V$. A detailed discussion of the choice of hyperprior is provided in Appendix D.

### 2.4. Extension to the nonlinear setting

NARD can be naturally extended to address nonlinearity through kernel method. Specifically, we consider

$$Y = W\Phi(X) + \mathcal{E}, \quad \Phi(\cdot) \in \text{Polynomial, RBF, ...}$$

where $\Phi(X)$ represents a nonlinear feature mapping that transforms the input space into a higher-dimensional space, enabling more flexible modeling of complex relationships.

## 3. SEQUENTIAL NARD

### 3.1. Sequential update

Rather than pruning redundant or irrelevant features as in NARD, we employ a greedy approach that sequentially adds and removes features. The key difference is that the original NARD requires $\mathcal{O}(d^3)$ computations at the beginning of training, whereas the sequential update method begins with an almost empty model, consisting of only a few features, which significantly reduces the initial computational burden. We adopt a fast sequential optimization method to efficiently update the hyperparameters inspired by (Faul & Tipping, 2001; Tipping & Faul, 2003; Ament & Gomes, 2021).

### 3.2. Fast optimization of evidence

To perform a sequential update on $V$ and $K$, we separate out the contribution of a single prior parameter $\alpha_i$ from the MLF $P(Y|X,V,K)$.

Thus, by rewriting $C$ as $C = C_{\setminus i} + \alpha_i^{-1}\varphi_i\varphi_i^\top$, and using determinant and inverse lemma of matrix, the logarithm of MLF can be explicitly decomposed into two parts, one part denoted by $\mathcal{L}(\alpha_{\setminus i})$, that does not depend on $\alpha_i$ and another that does, i.e.,

$$\mathcal{L}(\alpha) = m\left[\ln \alpha_i - \ln(\alpha_i + s_i)\right] + \frac{\mathbf{Tr}(q_i q_i^\top V^{-1})}{\alpha_i + s_i} \\ + \mathcal{L}(\alpha_{\setminus i}), \quad (19)$$

where $\varphi_i \in \mathbb{R}^N$ denotes the $i$-th row of $X$, $q_i = YC_{\setminus i}^{-1}\varphi_i$ and $s_i = \varphi_i^\top C_{\setminus i}^{-1}\varphi_i$.

**Theorem 3.1.** *Denote $\eta_i := \mathbf{Tr}(q_i q_i^\top V^{-1}) - ms_i$, then the global maximum of $\mathcal{L}(\alpha)$ with respect to $\alpha_i$ is*

$$\alpha_i = \begin{cases} \frac{ms_i^2}{\eta_i} & , \quad \eta_i > 0; \\ \infty & , \quad \eta_i \leq 0. \end{cases} \quad (20)$$

The detailed proof of Theorem 3.1 can be found in Appendix C.1. Furthermore, to simplify the maintenance and update of $s_i$ and $q_i$, we can exploit the following relations

$$q_i = \frac{\alpha_i Q_i}{\alpha_i - S_i}, \quad s_i = \frac{\alpha_i S_i}{\alpha_i - S_i}, \quad (21)$$

where we define

$$Q_i := YC_{\mathcal{A}}^{-1}\varphi_i, \quad S_i := \varphi_i^\top C_{\mathcal{A}}^{-1}\varphi_i. \quad (22)$$

Here $\mathcal{A}$ is the active subset of features, with

$$C_{\mathcal{A}}^{-1} := (I + X_{\mathcal{A}}^\top K_{\mathcal{A}}^{-1} X_{\mathcal{A}})^{-1}. \quad (23)$$

$X_{\mathcal{A}}$ and $K_{\mathcal{A}}$ are the sub-matrices of $X$ and $K$ corresponding to $A$. We denote $S_{xx}^{\mathcal{A}} := X_{\mathcal{A}} X_{\mathcal{A}}^\top + K_{\mathcal{A}}$. Using Woodbury identity, we can write

$$Q_i = Y\varphi_i - YX_{\mathcal{A}}^\top (S_{xx}^{\mathcal{A}})^{-1} X_{\mathcal{A}}\varphi_i, \\ S_i = \varphi_i^\top \varphi_i - \varphi_i^\top X_{\mathcal{A}}^\top (S_{xx}^{\mathcal{A}})^{-1} X_{\mathcal{A}}\varphi_i. \quad (24)$$

It can be shown that the computation in Eq. (24) involves only those features in the active set $\mathcal{A}$ that correspond to finite hyperparameters $\alpha_i$. Let $p \ll d$ denote the final number of features in the model, the computational complexity per iteration is $\mathcal{O}(p^3)$.

It is easy to verify that when $\alpha_i \to \infty$, both the prior and posterior of $W$ exhibit a very high density around $W_i = \vec{0}$. This indicates that when $\alpha_i \to \infty$, we can remove the $i$-th feature of $X$. See details in Appendix C.2.

### 3.3. Algorithm summary

Algorithm 2 summarizes the proposed Sequential NARD. We begin with a model containing only a few features, which

---

**Algorithm 2** Sequential NARD

**Input:** Input data $X, Y, \epsilon, T$
**Output:** Estimated $\alpha, V, V^{-1}$
1: Initialization: model with a few features, set $\alpha$ elements, $V \leftarrow \hat{V}_{\text{MLE}}, \Delta L(\alpha) \leftarrow \infty, t \leftarrow 0$.
2: **while** $\Delta L(\alpha) > \epsilon$ and $t < T$ **do**
3:    $t \leftarrow t + 1$.
4:    **select** $i \in \{1, 2, \ldots, d\}$ randomly.
5:    Calculate temporary $Q_i, q_i, S_i, s_i, \eta_i, \alpha_i$.
6:    **if** $\alpha_i < \infty$, i.e., $\varphi_i$ is in the model **then**
7:       **if** $\eta_i > 0$ **then**
8:          $\alpha_i \leftarrow \frac{ms_i^2}{\eta_i}$.
9:       **else**
10:          $\alpha_i \leftarrow \infty$, Delete $\varphi_i$ from $X$.
11:       **end if**
12:    **else**
13:       **if** $\eta_i > 0$ **then**
14:          $\alpha_i \leftarrow \frac{ms_i^2}{\eta_i}$, Add $\varphi_i$ to $X$.
15:       **else**
16:          Skip $\varphi_i$, continue.
17:       **end if**
18:    **end if**
19:    Calculate $K_{\mathcal{A}}, K_{\mathcal{A}}^{-1}, C_{\mathcal{A}}, C_{\mathcal{A}}^{-1}$.
20:    $\hat{V}_{\text{MLE}} \leftarrow \frac{Y^\top C_{\mathcal{A}}^{-1} Y}{N}$.
21:    $V, V^{-1} \leftarrow \mathbf{glasso}(\hat{V}_{\text{MLE}}, \lambda)$.
22:    Calculate $L, \Delta L(\alpha)_{\text{new}}$.
23:    **if** $\Delta L(\alpha)_{\text{new}} > 0$ **then**
24:       Update $X, Q, q, S, s, \eta, \alpha, L, K_{\mathcal{A}}, C_{\mathcal{A}}, V, V^{-1}$.
25:    **else**
26:       Undo all changes in this iteration.
27:       Continue
28:    **end if**
29: **end while**

---

means that the corresponding $\alpha$ elements are not $\infty$. Next, we randomly select $i \in \{1, 2, \ldots, d\}$, calculate $\eta_i$ and $\alpha_i$, and update $V$ using graphical lasso. Based on the values of $\eta_i$ and $\alpha_i$, and whether the $i$-th feature is currently included in the model, we decide whether to add it ($\alpha_i \leftarrow \frac{ms_i^2}{\eta_i}$), re-estimate it ($\alpha_i \leftarrow \frac{ms_i^2}{\eta_i}$), or delete it ($\alpha_i \leftarrow \infty$). Afterward, we compute the new MLF, $\mathcal{L}(\alpha)_{\text{new}}$. If it increases, we retain the change; otherwise, we undo it. This process is repeated until $\mathcal{L}(\alpha)$ converges.

## 4. SURROGATE NARD

### 4.1. Overview of surrogate function method

An alternative approach to reducing computational complexity is to introduce a surrogate function. More specifically, we approximate $p(Y|X, V, K, W)$ with a surrogate function

$\hat{p}(Y|X, V, K, W, W')$, satisfying

$$p(Y|X, V, K, W) = \max_{W'} \hat{p}(Y|X, V, K, W, W'), \quad (25)$$

which establishes a tight lower bound on the original likelihood function. In other words, our objective becomes to maximize the lower bound of the MLF. In fact, this is a common technique in optimization and variational inference (Hunter & Lange, 2004; Kingma & Welling, 2014; Sun et al., 2016). In particular, the most computationally expensive step of the original NARD is matrix inversion. To mitigate this, a lower bound is strategically chosen to approximate the matrices involved in the inversion as diagonal matrices. This approximation eliminates the bottleneck caused by matrix inversion, resulting in a substantial reduction in overall computational complexity.

### 4.2. Lower bound of the MLF

**Lemma 4.1.** *(Boyd & Vandenberghe, 2004) Suppose $f(X)$ is a function $f : \mathbb{R}^{n \times n} \to \mathbb{R}$, the first-order Taylor approximation with trace as inner product is*

$$f(X + V) \approx f(X) + \boldsymbol{Tr}(\nabla f(X)^\top V). \quad (26)$$

**Lemma 4.2.** *Let $f : \mathbb{R}^{n \times n} \to \mathbb{R}$ be a continuously differentiable function with Lipschitz continuous gradient and Lipschitz constant $L$. Then, for any $U, V \in \mathbb{R}^{n \times n}$,*

$$|f(U) - f(V) - \boldsymbol{Tr}(\nabla f(V)^\top (U - V))| \le \frac{L}{2} \|U - V\|^2. \quad (27)$$

Denote

$$\begin{aligned} R(W, W') &= (Y - W'X)(Y - W'X)^\top \\ &\quad + 2(W - W')X(W'X - Y)^\top \quad (28) \\ &\quad + \rho(W - W')(W - W')^\top, \end{aligned}$$

where $\rho \in \mathbb{R}$ denotes the largest eigenvalue of $XX^\top$, we have the following lemma:

**Lemma 4.3.** *Let $g(W) = (Y - WX)(Y - WX)^\top$, then $\boldsymbol{Tr}(g(W)) \le \boldsymbol{Tr}(R(W, W'))$.*

*Proof.* We begin by analyzing the Lipschitz constant $L$ for the function $g(W)$. First, we compute the gradient of $g(W)$ with respect to $W$:

$$\frac{\partial g(W)}{\partial W} = -2(Y - WX)X^\top. \quad (29)$$

According to the definition of Lipschitz continuous gradient, the following inequality holds:

$$\begin{aligned} &\left\| \frac{\partial g(W)}{\partial W} - \frac{\partial g(W')}{\partial W'} \right\| \le L\|W - W'\| \\ \Rightarrow\ & \|2(W - W')XX^\top\| \le L\|W - W'\| \quad (30) \\ \Rightarrow\ & L = 2\|XX^\top\| = 2\rho. \end{aligned}$$

Thus, we derive the desired inequality by applying Lemma 4.2, as shown below:

$$\begin{aligned} \boldsymbol{Tr}[g(W)] \le{}& \boldsymbol{Tr}[g(W')] + \boldsymbol{Tr}[\nabla g(W')^\top (W - W')] \\ &+ \frac{L}{2} \boldsymbol{Tr}[(W - W')(W - W')^\top]. \end{aligned}$$
$$(31)$$
$$\square$$

Revisiting the MLF in Eq. (25), we obtain the approximate posterior density of $W$ by substituting $p(Y|X, V, K, W)$ with its lower bound $\hat{p}(Y|X, V, K, W, W')$ via Bayesian rules as follows:

$$\begin{aligned} p(W|V, K) &\approx \frac{\hat{p}(Y|X, V, W, W')p(W|V, K)}{\int \hat{p}(Y|X, V, W, W')p(W|V, K)dw} \quad (32) \\ &= \mathcal{MN}(\mu, V, S_{xx}). \end{aligned}$$

with

$$\begin{aligned} \mu &= \rho W' - W'XX^\top + YX^\top, \\ S_{xx} &= \rho I + K. \end{aligned} \quad (33)$$

By substituting Eq. (32) into the marginal likelihood expression, we arrive at the following formulation:

**Proposition 4.4.** *The new marginal likelihood function, up to a constant, is*

$$\begin{aligned} \mathcal{L} &= \ln p(Y|X, V, K, W, W') \\ &\propto m \ln |C| + N \ln |V| + \boldsymbol{Tr}(V^{-1} R(W, W')) \quad (34) \\ &\quad + \boldsymbol{Tr}(V^{-1} W K W^\top). \end{aligned}$$

This reformulation leads to a new objective function involving 4 sets of variables $W, W', V, K$. Direct joint optimization of these variables is intractable due to their interdependence. Accordingly, we adopt the BCD method to optimize the negative logarithm of MLF, i.e.,

$$(W^{(k)}, W'^{(k)}, V^{(k)}, K^{(k)}) \in \arg\min \mathcal{L}(W, W', V, K). \quad (35)$$

Specifically, the BCD method is utilized to alternatively solve Eq. (35) as follows:

$$\begin{aligned} W^{(k)} &\in \arg\min_{W} \mathcal{L}(W, W'^{(k-1)}, V^{(k-1)}, K^{(k-1)}), \\ W'^{(k)} &\in \arg\min_{W'} \mathcal{L}(W^{(k)}, W', V^{(k-1)}, K^{(k-1)}), \\ V^{(k)} &\in \arg\min_{V} \mathcal{L}(W^{(k)}, W'^{(k)}, V, K^{(k-1)}), \\ K^{(k)} &\in \arg\min_{K} \mathcal{L}(W^{(k)}, W'^{(k)}, V^{(k)}, K). \end{aligned} \quad (36)$$

The optimal value of $W^{(k)}, W'^{(k)}, V^{(k)}, K^{(k)}$ can be obtained by setting its gradient to zero respectively, leading to

the following update schemes:

$$S_{xx}^{(k)} = K^{(k-1)} + \rho I, \tag{37}$$

$$W^{(k)} = \left[\rho W'^{(k-1)} - W'^{(k-1)} X X^\top + Y X^\top\right](S_{xx}^{(k)})^{-1}, \tag{38}$$

$$W'^{(k)} = W^{(k)}, \tag{39}$$

$$V^{(k)} = \frac{R(W^{(k)}, W'^{(k)}) + W^{(k)} K^{(k-1)} (W^{(k)})^\top}{N}, \tag{40}$$

$$V^{(k)}, \quad (V^{(k)})^{-1} \leftarrow \mathbf{glasso}(V^{(k)}, \lambda), \tag{41}$$

$$K^{(k)} = \frac{m}{\mathbf{diag}[(W^{(k)})^\top (V^{(k)})^{-1} W^{(k)} + m(S_{xx}^{(k)})^{-1}]}. \tag{42}$$

### 4.3. Algorithm summary

---
**Algorithm 3** Surrogate NARD

---
**Input:** Input data $X, Y, \epsilon$
**Output:** Estimated $\alpha, V, W$
1: Initialize $\alpha$ elements, $V \leftarrow \hat{V}_{\mathrm{MLE}}$, $W^{(0)} \leftarrow Y X^\top (K + X X^\top)^{-1}$.
2: **for** $k = 1$ **to** $K$ **do**
3:      Perform equations (37), (38), (39), (40), (41), (42).
4:      **if** $\|W^{(k)} - W^{(k-1)}\|_F \leq \epsilon$ **then**
5:          **Break**.
6:      **end if**
7: **end for**

---

Algorithm 3 presents a comprehensive overview of Surrogate NARD. Since $S_{xx}$ is a diagonal matrix, the update in (37) involves only the inversion of a diagonal matrix, which has a computational complexity of $\mathcal{O}(d)$.

## 5. Hybrid NARD

This section presents a hybrid method that integrates Sequential NARD and Surrogate NARD to further reduce computational costs. Specifically, we utilize a sequential update approach for the iterative steps of Surrogate NARD. This approach initiates by assessing the relevance of a new feature to determine its inclusion, akin to the Sequential NARD framework.

Once a feature is deemed relevant and included, we employ Surrogate NARD for subsequent calculations of the matrices $V, W$. This strategy eliminates the need for the costly matrix inversion operations typically required in full NARD implementations. By combining the feature selection process of Sequential NARD with the efficient computations of Surrogate NARD, we achieve a significantly lower computational complexity. The resulting method not only streamlines the iterative process but also ensures that we maintain comparable performance while handling high-dimensional data.

Hybrid NARD retains the core structure and principles of Sequential NARD, with key variables such as $Q, q, S, s,$ and $\eta$ preserved, and updates are performed only when the MLF increases. However, the key difference is that Hybrid NARD requires the computation of $\rho$ and the execution of steps (37), (38), (39), (40), (41) and (42) at each iteration, rather than relying on the updates specified in Sequential NARD. Consequently, $W$ is updated in every iteration of Hybrid NARD, whereas in Sequential NARD, $W$ is calculated only after the final values of $X, K$ and $V$ have been determined.

## 6. EXPERIMENTS

In this section, we evaluate the performance of the proposed methods on synthetic data using True Positive Rate (TPR), defined as $\mathrm{TPR} = \frac{\mathrm{TP}}{\mathrm{TP+FN}}$, and False Positive Rate (FPR), defined as $\mathrm{FPR} = \frac{\mathrm{FP}}{\mathrm{TN+FP}}$, where TP, FP, TN and FN represent true positives, false positives, true negatives, and false negatives, respectively. We run each experiment 10 times with different random seeds and report the average. For Aging phenotype data, where true labels are unavailable, we use the Jaccard index to measure the overlap of biological associations identified by different algorithms. We also highlight the efficiency of the NARD and its variants. On the TCGA data, we focus on the effectiveness of NARD in identifying biological associations. In addition, we conducted experiments on financial and air quality datasets to further demonstrate the versatility of our method; the detailed experimental results are provided in Appendix F.2. All experiments were performed on 32 Intel(R) Xeon(R) Platinum CPUs.

To select the optimal $\lambda$, we employ a 5-fold cross-validation procedure. The dataset is partitioned into 5 disjoint subsets, and in each iteration, 1 subset is held out as the validation set while the remaining 4 subsets are used for model estimation. The objective function for selecting $\lambda_{\mathrm{glasso}}$ is defined as:

$$\lambda_{\mathrm{glasso}} = \arg\min_\lambda \sum_{l=1}^{5} \left[\mathbf{Tr}(\tilde{V}_l \Omega_{-l}) - \log|\Omega_{-l}| + \lambda \sum_{i \neq j} |\omega_{ij}|\right]. \tag{43}$$

Here $\tilde{V}_l$ is the empirical covariance estimator computed from the training data excluding the $l$-th fold, and $\Omega_{-l}$ is the estimated precision matrix based on this subset. The log-likelihood is computed for each fold, and the $\lambda$ that maximizes the cross-validated log-likelihood is chosen. A grid search is performed over a range of candidate values for $\lambda$, and the value that yields the best performance across all folds is selected for the final model and evaluation.

### 6.1. Synthetic datasets

The synthetic data are generated as follows. The covariance matrix $V$ is constructed based on a graph generated by an

Erdős-Rényi random graph (Erdős & Rényi, 1959) with a sparsity parameter $p$. The entries of the precision matrix are sampled from a uniform distribution, after which the matrix is symmetrized and adjusted to ensure positive definiteness by controlling its minimum eigenvalue. The matrix $W$ is generated with a different sparsity level, and its non-zero elements are drawn from a uniform distribution. Both the data matrix $X$ and the error term $E$ are sampled from their corresponding multivariate normal distributions, and the outcome $Y$ is computed as $Y = WX + E$.

We choose two representative categories of baseline methods: MRCE[1] (Rothman et al., 2010) and CAPME[2] (Cai et al., 2013) as frequency-based approaches, and HS-GHS[3] (Li et al., 2021) and JRNS [4] (Samanta et al., 2022) as Bayesian sampling-based algorithms.

Table 1. Performance comparison of various methods. ($p = 0.1$)

| METHOD | $d$ | $m$ | $N$ | TPR | FPR | TIME (TOTAL) |
|---|---|---|---|---|---|---|
| MRCE | 5000 | 1500 | 1500 | 0.9083 | 0.0072 | 53 |
| CAPME | 5000 | 1500 | 1500 | 0.8972 | 0.0124 | 52 |
| HS-GHS | 5000 | 1500 | 1500 | 0.9463 | 0.0033 | >3000 |
| JRNS | 5000 | 1500 | 1500 | 0.9485 | 0.0037 | >3000 |
| NARD | 5000 | 1500 | 1500 | 0.9483 | 0.0062 | 49 |
| SEQUENTIAL NARD | 5000 | 1500 | 1500 | 0.9459 | 0.0067 | 35 |
| SURROGATE NARD | 5000 | 1500 | 1500 | 0.9462 | 0.0072 | 31 |
| HYBRID NARD | 5000 | 1500 | 1500 | 0.9471 | 0.0068 | 23 |

Table 1 presents the results with $N = 1500$, $m = 1500$, and $d = 5000$, highlighting estimation performance and CPU times for all methods. NARD and its variants perform similarly to HS-GHS, slightly outperforming MRCE in estimation, while demonstrating better computational efficiency. Among the methods, HS-GHS is the most time-consuming, while NARD-based approaches maintain relatively high efficiency, with performance gains becoming more pronounced as the dataset size increases. Table 2 compares performance under different combinations of $d$, $m$, and $N$, with a focus on time performance relative to MRCE.

Figure 1 illustrates the running times of NARD and Surrogate NARD for varying $d$ values ($d \in \{1500, 5000, 10000, 15000, 20000\}$ with $N = 1500$ and $m = 1500$). The results show that while the computational time of NARD increases significantly as $d$ grows, Surrogate NARD exhibits a lower time overhead. These trends are consistent with the theoretical complexity of the methods.

[1] https://cran.r-project.org/web/packages/MRCE/index.html
[2] http://www-stat.wharton.upenn.edu/~tcai/paper/Softwares/capme_1.3.tar.gz
[3] https://github.com/liyf1988/HS_GHS
[4] https://github.com/srijata06/JRNS_Stepwise

Table 2. Impact of data size on performance. ($p = 0.02$)

| METHOD | $d$ | $m$ | $N$ | TPR | FPR | TIME (1 STEP) |
|---|---|---|---|---|---|---|
| MRCE | 1000 | 1000 | 5000 | 0.9802 | 0.0313 | 3.4 |
| NARD | 1000 | 1000 | 5000 | 0.9825 | 0.0274 | 3.5 |
| SURROGATE NARD | 1000 | 1000 | 5000 | 0.9818 | 0.0281 | 2.8 |
| MRCE | 2000 | 2000 | 10000 | 0.9797 | 0.0293 | 8.7 |
| NARD | 2000 | 2000 | 10000 | 0.9864 | 0.0205 | 8.7 |
| SURROGATE NARD | 2000 | 2000 | 10000 | 0.9838 | 0.0223 | 6.0 |
| MRCE | 5000 | 2000 | 20000 | 0.9698 | 0.0411 | 19.2 |
| NARD | 5000 | 2000 | 20000 | 0.9732 | 0.0323 | 17.4 |
| SURROGATE NARD | 5000 | 2000 | 20000 | 0.9732 | 0.0379 | 13.0 |

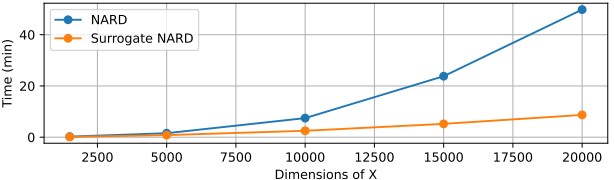

Figure 1. Time comparison for different methods.

## 6.2. Baselines

We further evaluate the scalability of NARD on larger datasets. As shown in Table 3, the experimental results align with the theoretical complexity, demonstrating that as the data size increases, the time per iteration for NARD grows in a manner consistent with its expected computational behavior. The Surrogate NARD approach consistently outperforms NARD, showing substantial improvements in computational efficiency, especially at larger scales.

Table 3. Average time per iteration. ($N = 20000, m = 2000$)

| $d$ | MRCE | NARD | SURROGATE NARD |
|---|---|---|---|
| 500 | 2.4 | 2.1 | 1.56 |
| 1000 | 2.5 | 2.4 | 1.96 |
| 2000 | 3.6 | 3.4 | 2.2 |
| 5000 | 12.2 | 10.7 | 3.7 |
| 10000 | 33.4 | 30.8 | 8.9 |
| 15000 | 77.5 | 70.9 | 20.8 |
| 20000 | 201.9 | 168.6 | 33.7 |
| 30000 | 421.3 | 376.7 | 64.7 |

## 6.3. Aging phenotype data

We use data from a cohort of 1022 healthy individuals to construct a phenotypic network closely related to aging. In total, we identify 5641 phenotypes that exhibited a significant Pearson correlation with age, including 1522 macro phenotypes and 4119 molecular phenotypes.
Figure 2 illustrates the phenotypic network. Notably, features within the range of 280-370 exhibited a pronounced block structure, which corresponds to phenotypic data de-

Table 4. Associations under different algorithms.

| METHOD | MRCE | HS-GHS | NARD | NARD VARIANTS | | |
|---|---|---|---|---|---|---|
| | | | | SEQUENTIAL | SURROGATE | HYBRID |
| # OF ASSOCIATION | 15330 | 14983 | 15101 | 15014 | 15062 | 15039 |
| JACCARD INDEX | 0.979 | - | 0.988 | 0.988 | 0.989 | 0.989 |

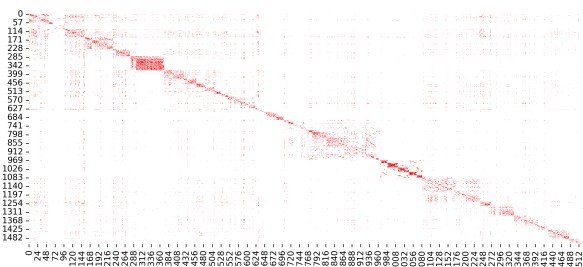

Figure 2. Phenotype network in aging.

rived from the same tissue type. This structural coherence underscores the biological relevance of the identified phenotypes and suggests that similar traits may arise from shared underlying biological mechanisms. Furthermore, various algorithms demonstrate consistent performance on the dataset, with the Jaccard index for phenotype associations surpassing 98.5%. In terms of computational efficiency, the NARD method is the slowest, taking approximately 24 minutes to complete, while the sequential NARD approach take approximately 14 minutes.

## 6.4. TCGA cancer data

To evaluate the effectiveness of NARD, we analyze data from The Cancer Genome Atlas (TCGA) (Weinstein et al., 2013) across seven tumor types: colon adenocarcinoma (COAD), lung adenocarcinoma (LUAD), lung squamous cell carcinoma (LUSC), ovarian serous cystadenocarcinoma (OV), rectum adenocarcinoma (READ), skin cutaneous melanoma (SKCM), and uterine corpus endometrial carcinoma (UCEC). Each cancer type dataset includes mRNA expression profiles and RPPA-based proteomic data, reflecting the biological relationship where mRNA is translated into proteins. We choose 10 key signaling pathways based on recent studies (Akbani et al., 2014; Cherniack et al., 2017; Li et al., 2017) of RPPA-based proteomic profiling across various tumor types.

Figure 3 visualizes the UpSet plots (Lex et al., 2014) for 7 cancers in 10 pathways. The translational effects exhibit heterogeneity across cancer types, reflecting the expected biological differences. Figure 4 presents the protein association network for COAD, highlighting interactions between proteins within the same pathway, as well as significant cross-talk between different pathways. The network plots for the other cancer types are included in the Appendix F.4.

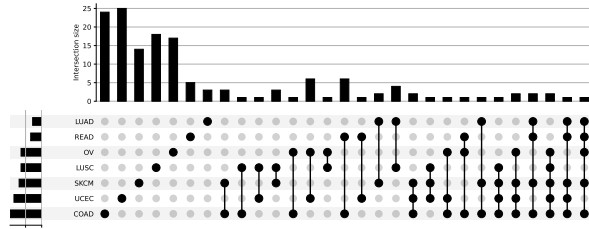

Figure 3. UpSet plots illustrating the relationships among 10 pathways across 7 cancer types.

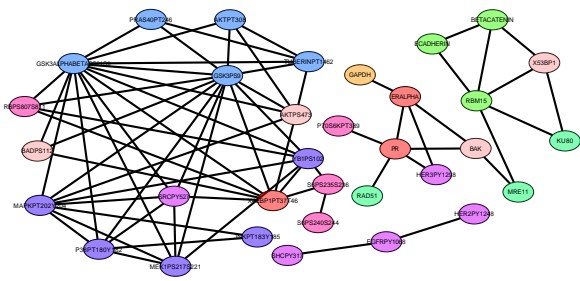

Figure 4. Protein network of COAD. Different colors represent different pathways.

Sparsity-inducing priors like ARD enhance interpretability in biological applications, such as TCGA cancer data, by identifying key features. In our analysis across 7 tumor types, ARD highlighted important genes and proteins linked to signaling pathways. In Figure 3, sparsity revealed consistent pathways across cancer types, exposing cancer-specific translational effects. In Figure 4, for COAD, sparsity highlighted critical protein interactions within pathways and cross-talk between them, aiding biological interpretation. In COAD, the PI3K/AKT pathway was highlighted by the interaction between GSK3ALPHABETAPS21S9 and AKTPS473 (Li et al., 2024). This association indicates a key regulatory role in tumor growth and survival. The AKT signaling axis, activated by various upstream kinases like GSK3, has been implicated in colon cancer progression, making it a valuable target for further investigation and therapeutic development (Zhang et al., 2021; Yao et al., 2020).

## 7. Conclusion

In this paper, we introduce the NARD framework and propose three variants to alleviate its computational burden, significantly reducing the cost while maintaining performance. Experimental results confirm the effectiveness and efficiency of our methods across diverse domains. While the original model assumes linear relationships, we have demonstrated that NARD can be readily extended to nonlinear settings via kernel-based techniques, further broadening its applicability.

## Acknowledgements

Authors acknowledge the constructive feedback of reviewers and the work of ICML'25 program and area chairs. This work was supported by the National Natural Science Foundation of China (Grant No. 82394432 and 92249302), and the Shanghai Municipal Science and Technology Major Project (Grant No. 2023SHZDZX02). Technical support was provided by the Human Phenome Data Center of Fudan University and the CFFF platform of Fudan University. This research was conducted during an internship at the Shanghai Academy of Artificial Intelligence for Science.

## Impact Statement

This paper presents work whose goal is to advance the field of Machine Learning. There are many potential societal consequences of our work, none which we feel must be specifically highlighted here.

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

## A. Matrix Computation

### A.1. Trace and determinant

$$
\begin{aligned}
\mathbf{Tr}(ABC) &= \mathbf{Tr}(CAB) = \mathbf{Tr}(BCA), \\
\left|I_N + AB^\top\right| &= \left|I_M + A^\top B\right|, \qquad \forall A, B \in \mathbb{R}^{N \times M}.
\end{aligned}
\tag{44}
$$

### A.2. WoodBury identity

$$
(A + BD^{-1}C)^{-1} = A^{-1} - A^{-1}B(D + CA^{-1}B)^{-1}CA^{-1}.
\tag{45}
$$

### A.3. Derivatives of matrix-variate function

$$
\begin{aligned}
\frac{\partial \mathbf{Tr}(X^\top BXC)}{\partial X} &= BXC + B^\top XC^\top, \\
\frac{\partial \mathbf{Tr}(AXB)}{\partial X} &= A^\top B^\top, \\
\frac{\partial \ln|X|}{\partial X} &= (X^{-1})^\top.
\end{aligned}
\tag{46}
$$

### A.4. Contribution of $\alpha_i$

Let $C = I + X^\top K^{-1} X = C_{\backslash i} + \alpha_i^{-1} \varphi_i \varphi_i^\top$, then we have

$$
\begin{aligned}
|C| &= |C_{\backslash i}||1 + \alpha_i^{-1} \varphi_i^\top C_{\backslash i}^{-1} \varphi_i|, \\
C^{-1} &= C_{\backslash i}^{-1} - \frac{C_{\backslash i}^{-1} \varphi_i \varphi_i^\top C_{\backslash i}^{-1}}{\alpha_i + \varphi_i^\top C_{\backslash i}^{-1} \varphi_i}.
\end{aligned}
\tag{47}
$$

## B. Derivations in Detail

### B.1. Details of Eq. (9)

$\ln p(Y, W|X, V, K)$ is the joint distribution of $Y$ and $W$. By multiplying the likelihood function with the conjugate prior and omitting the constant, we get the kernel of $\ln p(Y, W|X, V, K)$.

Recall that the distribution of $Y$ given $X$ under the model is

$$
\begin{aligned}
p(Y|X; W, \epsilon) &= \prod_i^N p(y_i|x_i, W, V) \\
&= \frac{1}{|2\pi V|^{N/2}} \exp\left(-\frac{1}{2} \sum_i (y_i - Wx_i)^\top V^{-1} (y_i - Wx_i)\right) \\
&= \frac{1}{|2\pi V|^{N/2}} \exp\left(-\frac{1}{2} \mathbf{Tr}\left(V^{-1}(Y - WX)(Y - WX)^\top\right)\right) \\
&= \frac{1}{|2\pi V|^{N/2}} \exp\left(-\frac{1}{2} \mathbf{Tr}\left(V^{-1}\left[WXX^\top W^\top - 2YX^\top W^\top + YY^\top\right]\right)\right).
\end{aligned}
\tag{48}
$$

and the conjugate prior of $W \sim \mathcal{MN}(0, V_{m \times m}, K_{d \times d}^{-1})$ is

$$
p(W; V, K^{-1}) = \frac{|K|^{m/2}}{(2\pi)^{md/2}|V|^{d/2}} \exp\left[-\frac{1}{2} \mathbf{Tr}(V^{-1}WKW^\top)\right].
\tag{49}
$$

Define

$$S_{xx} = XX^\top + K, S_{yy} = YY^\top, S_{yx} = YX^\top, S_{y|x} = S_{yy} - S_{yx}S_{xx}^{-1}S_{yx}^\top.$$

By multiplying the likelihood function $p(Y|W, X)$ with the conjugate prior of $W$, i.e., $p(W|V, K)$, and note that

$$\left(W - S_{yx}S_{xx}^{-1}\right) S_{xx} \left(W - S_{yx}S_{xx}^{-1}\right)^\top = W S_{xx} W^\top - 2 S_{yx} W^\top + S_{yx}S_{xx}^{-1}S_{yx}^\top. \tag{50}$$

We have

$$\ln p(Y, W|X, V, K) \propto \mathbf{Tr}\left[V^{-1}\left(W S_{xx} W^\top - 2 S_{yx} W^\top + S_{yy}\right)\right] \tag{51}$$

$$= \mathbf{Tr}\left[V^{-1}\left(W S_{xx} W^\top - 2 S_{yx} W^\top + S_{yx}S_{xx}^{-1}S_{yx} + S_{y|x}\right)\right] \tag{52}$$

$$= \mathbf{Tr}\left[V^{-1}\left(W - S_{yx}S_{xx}^{-1}\right) S_{xx} \left(W - S_{yx}S_{xx}^{-1}\right)^\top + V^{-1}S_{y|x}\right]. \tag{53}$$

## B.2. Details of Eq. (10)

In Bayesian analysis with conjugate priors, the normalization constants are often omitted during intermediate steps, and only the kernel of the distribution is considered. Afterward, if the form of the kernel matches a known distribution, the normalization factor can be reintroduced. In our case, since the matrix normal distribution is conjugate to the Gaussian likelihood, the posterior distribution of $W$ also follows a matrix normal distribution. Therefore, we only need to match the kernel of the posterior distribution with the known form of the matrix normal distribution to derive the posterior.

As a result, the first term in Eq. (9) can be decomposed into two parts: the first part corresponds to the kernel of the matrix normal distribution for $W$, and the second part corresponds to the kernel for $Y$.

Formally, we have $p(W|X, Y, V, K) \sim \mathcal{MN}(\mu, V, \Sigma)$, where $\mu = S_{yx}S_{xx}^{-1}$ and $\Sigma = S_{xx}$ are the posterior mean and column covariance of $W$, respectively. $C = I + X^\top K^{-1} X$ is the column covariance of $Y$.

For $Y$, $S_{y|x} = S_{yy} - S_{yx}S_{xx}^{-1}S_{yx}^\top = Y(I - X^\top S_{xx}^{-1} X)Y^T = YC^{-1}Y^\top$, where the third equality follows from WoodBury identity. Thus, we have $p(Y|X, V, K) \sim \mathcal{MN}(0, V, C)$.

## B.3. Details of Eq. (11)

$$P(Y|X, V, K) \sim \mathcal{MN}(0, V, C) = \frac{1}{(2\pi)^{mN/2}|C|^{m/2}|V|^{N/2}} \exp\left[-\frac{1}{2}\mathbf{Tr}(C^{-1}Y^\top V^{-1}Y)\right] \tag{54}$$

Hence,

$$\mathcal{L} = -\ln P(Y|X, V, K) = \frac{1}{2}\left[mN \ln 2\pi + m \ln|C| + N \ln|V|\right] + \frac{1}{2}\mathbf{Tr}(C^{-1}Y^\top V^{-1}Y) \tag{55}$$

$$\propto m \ln|C| + N \ln|V| + \mathbf{Tr}(Y^\top V^{-1}YC^{-1}) \tag{56}$$

## B.4. Details of Eq. (13)

The term $YC^{-1}Y^\top$ can be re-expressed by introducing the latent variable $W$ as follows:

$$YC^{-1}Y^\top = Y(I - X^\top S_{xx}^{-1} X)Y^\top \tag{57}$$

$$= YY^\top - YX^\top S_{xx}^{-1} XY^\top \tag{58}$$

$$= YY^\top - \mu XY^\top \tag{59}$$

$$= (Y - \mu X)(Y - \mu X)^\top + YX^\top\mu^\top - \mu XX^\top\mu^\top \tag{60}$$

$$= (Y - \mu X)(Y - \mu X)^\top + \mu S_{xx}\mu^\top - \mu XX^\top\mu^\top \tag{61}$$

$$= (Y - \mu X)(Y - \mu X)^\top + \mu(S_{xx} - XX^\top)\mu^\top \tag{62}$$

$$= (Y - \mu X)(Y - \mu X)^\top + \mu K\mu^\top \tag{63}$$

$$= \min_W (Y - WX)(Y - WX)^\top + WKW^\top, \tag{64}$$

where $\mu = YX^\top S_{xx}^{-1}$. The last equality represents the variational form. The value of $W$ that minimizes the expression is $W = \mu$, which can be derived by taking the derivative of the objective function $(Y - WX)(Y - WX)^\top + WKW^\top$ with respect to $W$ and setting it equal to zero.

## B.5. Details of Eq. (15)

Under flat prior, it can be shown that

$$\frac{\partial \mathcal{L}}{\partial V^{-1}} = -NV + YC^{-1}Y^\top = -NV + (Y - \mu X)(Y - \mu X)^\top + \mu K \mu^\top. \tag{65}$$

Hence,

$$V^{\text{new}} = \frac{(Y - \mu X)(Y - \mu X)^\top + \mu K \mu^\top}{N}. \tag{66}$$

## B.6. Details of Eq. (23)

Eq. (21), Eq. (22) and Eq. (23) is the standard quantity in sparse learning. To some extent, $s_i$ is called the **sparsity** and $q_i$ is known as the **quality** of $\phi$, The **sparsity** measures the extent to which basis function overlaps with the other basis vectors in the model, and the **quality** represents a measure of the alignment of the basis vector with the error between the training set values and the vector of predictions that would result from the model with the vector excluded.

Since features are sequentially added to the model in Sequential NARD, we only need to consider the features already presented in the model at each step. So in this case, we define $\mathcal{A}$ as the corresponding active subset of features. Therefore, we add a subscript $\mathcal{A}$ to the quantities in the analysis of this algorithm.

In Eq. (22), we have defined
$$Q_i = YC_{\mathcal{A}}^{-1}\varphi_i, \quad S_i = \varphi_i^\top C_{\mathcal{A}}^{-1}\varphi_i.$$

We have also defined $q_i = YC_{\backslash i}^{-1}\varphi_i$ and $s_i = \varphi_i^\top C_{\backslash i}^{-1}\varphi_i$. However, these two quantities involves $C_{\backslash i}^{-1}$ which leads to the high computational cost.

In the proof of Eq. (21) below. We omit the subscript $\mathcal{A}$ presented in $C_{\mathcal{A}}$ for readability.

$$S_i = \varphi_i^\top \left( C_{\backslash i}^{-1} - \frac{C_{\backslash i}^{-1}\varphi_i\varphi_i^\top C_{\backslash i}^{-1}}{\alpha_i + \varphi_i^\top C_{\backslash i}^{-1}\varphi_i} \right) \varphi_i \tag{67}$$

$$= \varphi_i^\top C_{\backslash i}^{-1}\varphi_i - \frac{(\varphi_i^\top C_{\backslash i}^{-1}\varphi_i)^2}{\alpha_i + s_i} \tag{68}$$

$$= s_i - \frac{s_i^2}{\alpha_i + s_i} \tag{69}$$

$$= \frac{\alpha_i s_i}{\alpha_i + s_i} \tag{70}$$

By rearranging it, we can easily get $s_i = \frac{\alpha_i S_i}{\alpha_i - S_i}$.

$$Q_i = YC^{-1}\varphi_i \tag{71}$$

$$= Y \left( C_{\backslash i}^{-1} - \frac{C_{\backslash i}^{-1}\varphi_i\varphi_i^\top C_{\backslash i}^{-1}}{\alpha_i + \varphi_i^\top C_{\backslash i}^{-1}\varphi_i} \right) \varphi_i \tag{72}$$

$$= YC_{\backslash i}^{-1}\varphi_i - \frac{(YC_{\backslash i}^{-1}\varphi_i)(\varphi_i^\top C_{\backslash i}^{-1}\varphi_i)}{\alpha_i + s_i} \tag{73}$$

$$= q_i - \frac{q_i s_i}{\alpha_i + s_i}. \tag{74}$$

By rearranging it, we get

$$q_i = \frac{(\alpha_i + s_i)Q_i}{\alpha_i} \tag{75}$$

$$= \frac{\left(\alpha_i + \frac{\alpha_i S_i}{\alpha_i - S_i}\right) Q_i}{\alpha_i} \tag{76}$$

$$= \frac{\alpha_i Q_i}{\alpha_i - S_i}. \tag{77}$$

## C. Details of Sequential NARD

### C.1. Proof of Theorem 3.1

**Theorem C.1.** *Denote* $\eta_i := \mathbf{Tr}(q_i q_i^\top V^{-1}) - m s_i$, *then the global maximum of* $L(\alpha)$ *with respect to* $\alpha_i$ *is*

$$\alpha_i = \begin{cases} \frac{ms_i^2}{\eta_i} & , \quad \eta_i > 0 \\ \infty & , \quad \eta_i \le 0 \end{cases} \tag{78}$$

*Proof.* The stationary points of the marginal likelihood with respect to $\alpha_i$ occur when

$$\frac{\partial \mathcal{L}(\alpha)}{\partial \alpha_i} = \frac{1}{2} \left[ \frac{m}{\alpha_i} - \frac{m}{\alpha_i + s_i} - \frac{\mathbf{Tr}(q_i q_i^\top V^{-1})}{(\alpha_i + s_i)^2} \right] = 0.$$

Then we have $\alpha_i^* = \frac{ms_i^2}{\eta_i}$. When $\eta_i > 0$, $\alpha_i^* > 0$, then this is a reasonable $\alpha_i$ value.

$$\begin{aligned} \frac{\partial^2 L}{\partial \alpha_i^2} &= \frac{1}{2} \left[ -\frac{m}{\alpha_i^2} + \frac{m}{(\alpha_i + s_i)^2} + \frac{2\mathbf{Tr}(q_i q_i^\top V^{-1})}{(\alpha_i + s_i)^3} \right] \\ &= \frac{1}{2} \left[ \frac{2\alpha_i^2 \eta_i - 3m\alpha_i s_i^2 - ms_i^3}{\alpha_i^2(\alpha_i + s_i)^3} \right]. \end{aligned} \tag{79}$$

Case 1. $\eta_i > 0$:
When $\alpha_i = \alpha_i^*$, the nominator is $2\frac{m^2 s_i^4}{\eta_i} - 3\frac{m^2 s_i^4}{\eta_i} - ms_i^3 = -\frac{m^2 s_i^4}{\eta_i} - ms_i^3$. So, it is negative.
Then, $\eta_i > \frac{-m^2 s_i^4}{ms_i^3} = -ms_i$ (already satisfied since here $\eta_i > 0$). Also, denominator is positive.
Hence, $\frac{\partial^2 L}{\partial \alpha_i^2}(\alpha_i^*) < 0$. Therefore, $\alpha_i^*$ is the maximum point.

Case 2. $\eta_i \le 0$:
Then $\alpha_i^* \notin \text{Dom}(L)$ since we need $\alpha_i > 0$ for all i, so $\alpha_i^*$ can not be the optimal point here.
Note that

$$\frac{\partial L}{\partial \alpha_i} = \frac{ms_i^2 + a\alpha_i s_i - \alpha_i \mathbf{Tr}(q_i q_i^\top V^{-1})}{(\alpha_i + s_i)^3} = \frac{ms_i^2 - \alpha_i \eta_i}{\alpha_i^2(\alpha_i + s_i)^3}.$$

We can observe that the nominator and the denominator is always positive. So $L$ is increasing with respect to $\alpha_i$. Hence, $\alpha_i \to \infty$ will make $L$ as large as possible. $\qquad\square$

Recall the Theorem 3.1, $s_i$ is called the sparsity and $q_i$ is known as the quality of $\varphi_i$, The sparsity measures the extent to which basis function overlaps with the other basis vectors in the model, and the quality represents a measure of the alignment of the basis vector with the error between the training set values and the vector of predictions that would result from the model with the vector excluded. The term $\eta_i = \text{Tr}(q_i q_i^\top V^{-1}) - ms_i$ actually measures the trade-off between the alignment quality of the basis vector and its sparsity in relation to the covariance structure. For $L(\alpha_i)$, when $\eta_i > 0$, the function exhibits an initial increase followed by a decrease, with the maximum value occurring at a stationary point. When $\eta_i \le 0$, the process is monotonically increasing, and the maximum value is asymptotically approached as $\alpha_i \to \infty$, consistent with the proof of Theorem 3.1. Furthermore, as $\alpha_i \to \infty$, the part of $L(\alpha_i)$ that depends on $\alpha_i$ diminishes, and $L(\alpha_i) = 0$ represents the situation where the corresponding feature can be pruned from the model.

## C.2. Analysis of prior

**Observation.** When $\alpha_i \to \infty$, $W_i \to \vec{0}$. $\mathbf{Tr}(KW^\top V^{-1}W) \geq 0$.

*Proof.* $K$ and $V$ are positive definite, so $K^{-1}$ and $V^{-1}$ are also positive definite. For all $x \in \mathbb{R}^m$, $x^\top W^\top V^{-1}Wx \geq 0$. Hence, $W^\top V^{-1}W$ is positive semidefinite.
Note, the trace of the product of two positive semidefinite matrices is nonnegative. So $\mathbf{Tr}(KW^\top V^{-1}W) \geq 0$.

$\square$

Prior

$$P(W|V,K) = \underbrace{\frac{|K|^{m/2}}{(2\pi)^{md/2}|V|^{d/2}}}_{A} \exp\left[-\frac{1}{2}\mathbf{Tr}(V^{-1}WKW^\top)\right].$$

When $\alpha_i \to \infty$ and fix $\alpha_{\backslash i}$, $|K| \to \infty$ then $A \to \infty$.

So,

$$\mathbf{Tr}(KW^\top V^{-1}W) \to \begin{cases} \infty & , W_i \neq \vec{0} \\ const & , W_i \to \vec{0} \end{cases}$$

Hence,

$$\exp\left[-\frac{1}{2}\mathbf{Tr}(KW^\top V^{-1}W)\right] \to \begin{cases} 0, & \text{if } W_i \neq \vec{0} \\ const, & \text{if } W_i \to \vec{0} \end{cases}$$

For a matrix $A \in \mathbb{R}^{a \times b}$ following a matrix normal distribution $\mathcal{MN}(M, U, Q)$, the density is

$$\frac{|U|^{-d/2}|Q|^{-m/2}}{(2\pi)^{md/2}} \exp\left\{-\frac{1}{2}\mathbf{Tr}\left[Q^{-1}(A-M)U^{-1}(A-M)^\top\right]\right\}, \tag{80}$$

where $M \in \mathbb{R}^{a \times b}$ is the expectation, $U \in \mathbb{R}^{a \times a}$ and $Q \in \mathbb{R}^{b \times b}$ are two positive definite matrices representing the covariance matrices for rows and columns of $A$ respectively.

# D. Hyperprior

## D.1. Hyperprior for $V$

**Flat prior.** It can be shown that

$$\frac{\partial \mathcal{L}}{\partial V^{-1}} = -NV + (Y - \mu X)(Y - \mu X)^\top + \mu K \mu^\top. \tag{81}$$

Hence,

$$V^{\text{new}} = \frac{(Y - \mu X)(Y - \mu X)^\top + \mu K \mu^\top}{N}. \tag{82}$$

**Inverse Wishart prior.** If $V$ follows a inverse Wishart distribution:

$$\begin{aligned} p(V) &\sim \mathcal{W}^{-1}(\Psi, \nu) \\ &\propto |\Psi|^{\frac{\nu}{2}}|V|^{-\frac{(\nu+m+1)}{2}} \exp\left(-\frac{1}{2}\mathbf{Tr}(V^{-1}\Psi)\right) \end{aligned} \tag{83}$$

The posterior for $V$ is still inverse Wishart:

$$p(V|X,Y) \sim \mathcal{W}^{-1}(S_{y|x} + \Psi, N + \nu), \tag{84}$$

and $Y$ is matrix-$\mathcal{T}$ distribution.

$$Y \sim \mathcal{T}(0, \Psi, C^{-1}, N + \nu). \tag{85}$$

Hence,

$$V^{\text{new}} = \frac{(Y - \mu X)(Y - \mu X)^\top + \mu K \mu^\top + \Psi}{N + \nu}. \tag{86}$$

### D.2. Hyperprior for $\alpha$

Here we consider different prior for $\alpha$.

**Flat prior.** If $\alpha$ follows the flat prior, i.e., $p(\alpha) = 1$, then

$$\frac{\partial \mathcal{L}}{\partial \ln(\alpha_i)} = m \left[ \alpha_i \left( S_{xx}^{-1} \right)_{ii} - 1 \right] + \alpha_i \left( \mu^\top V^{-1} \mu \right)_{ii}. \tag{87}$$

we have

$$\alpha_i^{\text{new}} = \frac{m}{m(S_{xx}^{-1})_{ii} + (\mu^\top V^{-1} \mu)_{ii}}. \tag{88}$$

**Gamma prior.** If $\alpha$ follows a Gamma prior Gamma$(a, b)$:

$$p(\alpha) = \prod_{i=1}^d \frac{b^a}{\Gamma(a)} \alpha_i^{a-1} e^{-b\alpha_i}. \tag{89}$$

where $a$ and $b$ are the shape and rate parameters of the Gamma distribution, respectively. Then

$$\frac{\partial \mathcal{L}}{\partial \ln(\alpha_i)} = \frac{m}{2} \left[ \alpha_i \left( S_{xx}^{-1} \right)_{ii} - 1 \right] + \frac{1}{2} \alpha_i \left( \mu^\top V^{-1} \mu \right)_{ii} + a - 1 - b\alpha_i, \tag{90}$$

therefore,

$$\alpha_i^{\text{new}} = \frac{m - 2a + 2}{m(S_{xx}^{-1})_{ii} + (\mu^\top V^{-1} \mu)_{ii} - 2b}. \tag{91}$$

## E. Relation with Other Work

We noticed that the model in the paper "An Iterative Min-Min Optimization Method for Sparse Bayesian Learning" (Wang et al., 2024) is designed for univariate regression, where the model is expressed as $y = Xw + \epsilon$, with $y, \epsilon \in \mathbb{R}^n$, $X \in \mathbb{R}^{n \times d}$, and $w \in \mathbb{R}^d$. In contrast, our approach focuses on a multivariate regression model. Therefore, direct comparison between the two models is not entirely appropriate unless we restrict the analysis to datasets with a single outcome variable.

To extend the Min-Min SBL methods from that paper to the multivariate case, several modifications are necessary. For example, we need to adjust the prior distribution for the regression coefficient matrix. Additionally, to ensure that the negative logarithm of the marginal likelihood function can be decomposed as shown in Eq 12 of Min-Min SBL, and to preserve the desirable properties of the Concave-Convex Procedure (CCCP) as discussed in Lemmas 2.1 and 2.2, we may need to introduce specific constraints into the model.

While extending the model may seem intuitively straightforward, the actual derivations are not trivial and require additional effort. For instance, when we introduce the covariance matrix $V$, all iterative formulas must be re-derived within the CCP framework. ARD, in this sense, is more of a framework, where the solution process includes the classical Mackay update, the EM update, and methods like Min-Min SBL. In our paper, we focus on improving the algorithmic complexity of the Mackay update step, while the extension offered by Min-Min SBL to ARD are orthogonal to the scope of our current work.

## F. Additional Experiments

### F.1. Extension to the nonlinear setting

Our method can be naturally extended to address nonlinearity through kernel method. Specifically, we consider the model

$$Y = W\Phi(X) + \mathcal{E}, \quad \Phi(\cdot) \in \text{Polynomial, RBF, ...}$$

*Table 5.* Associations under different algorithms with kernels.

| METHOD | MRCE | CAPME | HS-GHS | JRNS | NARD | NARD(POLYNOMIAL) | NARD(RBF) |
|---|---|---|---|---|---|---|---|
| # OF ASSOCIATION | 15330 | 15094 | 14983 | 15066 | 15101 | 15094 | 15072 |
| JACCARD INDEX | 0.979 | 0.977 | - | 0.988 | 0.988 | 0.990 | 0.989 |

where $\Phi(X)$ represents a nonlinear feature mapping that transforms the input space into a higher-dimensional space, allowing for more flexible modeling of complex relationships.

To explore this extension, we consider 2 different kernel functions: the polynomial kernel and the Gaussian (RBF) kernel. We evaluate the performance on a real-world aging phenotype data.

As shown in Table 5, our approach with polynomial and RBF kernels demonstrates competitive performance, achieving high Jaccard index values. The results are consistent with our expectation that kernel-based extensions allow the model to capture more complex, nonlinear relationships in the data, further validating the robustness and flexibility of our method.

## F.2. Dataset diversity

We also expanded our experiments to include 2 non-biological datasets: Kaggle's air quality dataset[5] and A-shares stock dataset.

For the air quality dataset, we performed data imputation and timestamp alignment, then analyzed the relationships among 11 key indicators. The results show a strong correlation between PM2.5 and humidity, supporting the environmental principle that higher humidity promotes the adhesion of fine particles, leading to increased PM2.5 levels. This aligns with previous studies on atmospheric dynamics.

For the A-shares dataset, we collect nearly 7 years of daily trading data and use the previous 5 days' information to predict the next day's opening price. This results in a dataset of 3032 stocks.

As shown in Table 6, our experiments show that Bayesian methods, such as HS-GHS and JRNS, were unable to complete calculations in 4 days, while our approach demonstrates excellent scalability. Analysis of the precision matrix reveals significant block structures, indicating that stocks from the same sector or industry tend to show similar trends in price movement. Through these 2 experimental datasets, we have demonstrated the effectiveness of our method in both environmental and financial domains. Since there is no ground truth, we used MRCE as a baseline algorithm for comparison. We reported the Jaccard index as a benchmark. Additionally, we presented the computational time to highlight the computational advantages of our method.

*Table 6.* Associations of A-shares stocks.($m = 3032, d = 60640, N = 1696$)

| METHOD | MRCE | CAPME | HS-GHS | JRNS | NARD | SEQUENTIAL NARD | SURROGATE NARD | HYBRID NARD |
|---|---|---|---|---|---|---|---|---|
| # OF ASSOCIATION | 97939 | 98335 | - | - | 99671 | 100309 | 99105 | 99475 |
| JACCARD INDEX | - | 0.869 | - | - | 0.881 | 0.891 | 0.893 | 0.890 |
| TIME PER ITERATION (SECOND) | 1200 | 1300 | - | - | 1000 | - | 255 | - |
| TIME ALL (H) | 17 | 16.5 | - | - | 14.5 | 8 | 5.5 | 3 |

## F.3. Pathway of TCGA cancer data

Table 7 shows the sample size $N$, the number of predictors $d$, and the number of response variables $m$ for the 7 datasets corresponding to each cancer type.

Table 8 presents a list of all pathways considered in the analysis of the TCGA cancer data in this study, along with their respective protein members.

*Table 7.* Datasets on seven different cancer types.

| CANCER | $N$ | $d$ | $m$ |
|--------|-----|-----|-----|
| READ | 121 | 73 | 76 |
| LUAD | 356 | 73 | 76 |
| COAD | 338 | 73 | 76 |
| LUSC | 309 | 73 | 86 |
| OV | 227 | 73 | 77 |
| SKCM | 333 | 73 | 76 |
| UCEC | 393 | 73 | 77 |

*Table 8.* Pathways and protein names.

| PATHWAY_SHORT | PROTEIN |
|---------------|---------|
| APOPTOSIS | BAK, BAX, BID, BIM, CASPASE7CLEAVEDD198, BADPS112, BCL2, BCLXL, CIAP |
| BREAST-REACTIVE | CAVEOLIN1, MYH11, RAB11, BETACATENIN, GAPDH, RBM15 |
| CELL-CYCLE | CDK1, CYCLINB1, CYCLINE1, CYCLINE2, P27PT157, P27PT198, PCNA, FOXM1 |
| CORE-REACTIVE | CAVEOLIN1, BETACATENIN, RBM15, ECADHERIN, CLAUDIN7 |
| DNA DAMAGE RESPONSE | 53BP1, ATM, BRCA2, CHK1PS345, CHK2PT68, KU80, MRE11, P53, RAD50, RAD51, XRCC1 |
| EMT | FIBRONECTIN, NCADHERIN, COLLAGENVI, CLAUDIN7, ECADHERIN, BETACATENIN, PAI1 |
| PI3K/AKT | AKTPS473, AKTPT308, GSK3ALPHABETAPS21S9, GSK3PS9, P27PT157, P27PT198, PRAS40PT246, TUBERINPT1462, INPP4B, PTEN |
| RAS/MAPK | ARAFPS299, CJUNPS73, CRAFPS338, JNKPT183Y185, MAPKPT202Y204, MEK1PS217S221, P38PT180Y182, P90RSKPT359S363, YB1PS102 |
| RTK | EGFRPY1068, EGFRPY1173, HER2PY1248, HER3PY1298, SHCPY317, SRCPY416, SRCPY527 |
| TSC/MTOR | 4EBP1PS65, 4EBP1PT37T46, 4EBP1PT70, P70S6KPT389, MTORPS2448, S6PS235S236, S6PS240S244, RBPS807S811 |

## F.4. Protein network for different cancer

Figure 5 and Figure 6 illustrate the protein networks for UCEC and LUAD, respectively. These networks highlight the interactions between key proteins associated with each cancer type. Figure 7 and Figure 8 show the networks for LUSC and OV, revealing distinct protein association patterns that may contribute to the unique characteristics of these cancers. Similarly, Figure 9 and Figure 10 present the protein networks for READ and SKCM, providing further insight into the molecular landscape of these cancer types.

## F.5. Further discussions

The Surrogate NARD sometimes demonstrate instability in its computations. This is related to numerical issues that arise during the iterative optimization process. Specifically, when working with large, high-dimensional real-world datasets, the precision matrix estimation can become unstable due to the ill-conditioning of the covariance matrix or the challenges associated with ensuring that the precision matrix remains positive definite throughout the iterations. This numerical instability is not related to gradient-based methods but rather stems from the nature of the data and the underlying optimization procedure. A potential solutions is that we can use a more robust initialization for the precision matrix.

---

[5]https://archive.ics.uci.edu/dataset/501/beijing+multi+site+air+quality+data

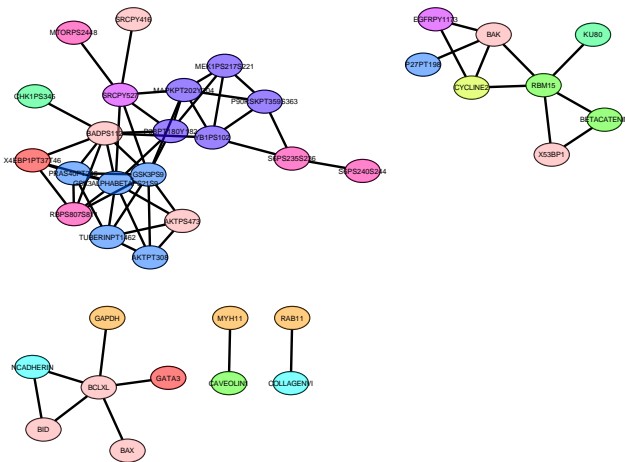

*Figure 5.* Protein network of UCEC.

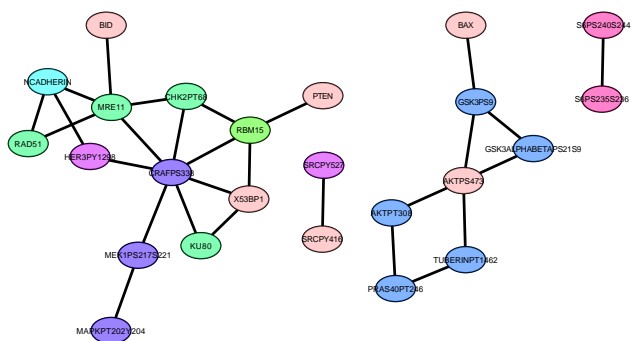

*Figure 6.* Protein network of LUAD.

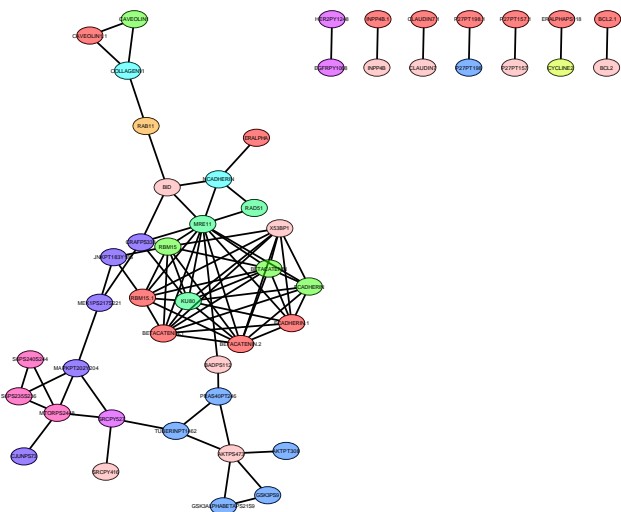

*Figure 7.* Protein network of LUSC.

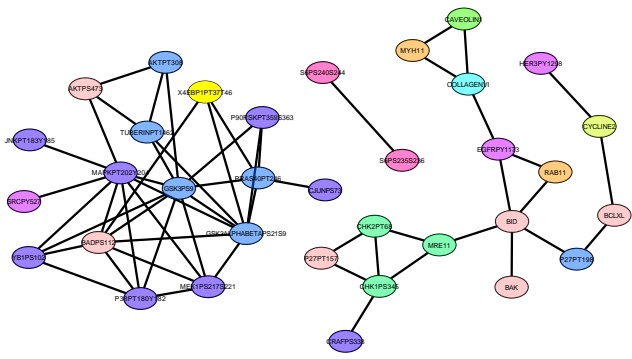

*Figure 8.* Protein network of OV.

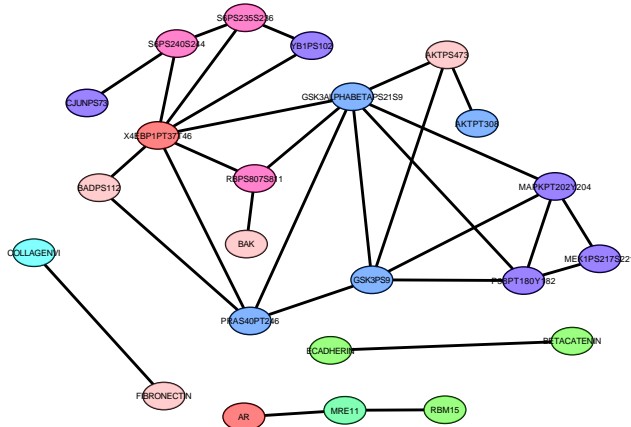

*Figure 9.* Protein network of READ.

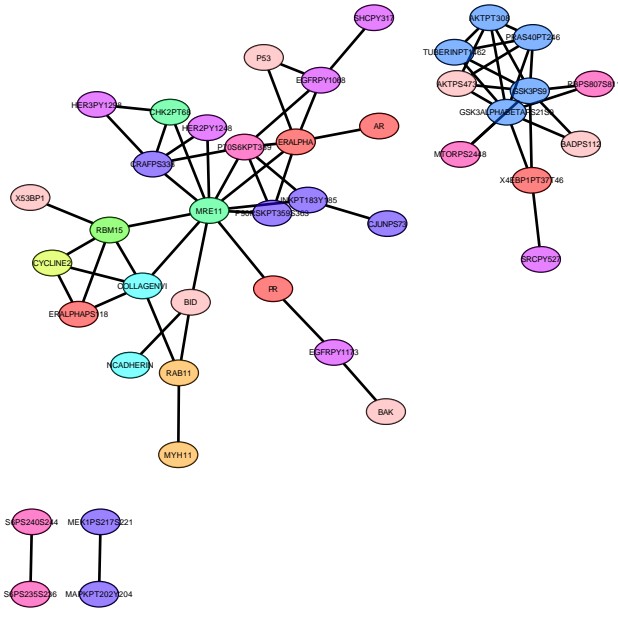

*Figure 10.* Protein network of SKCM.

