# OpenReview forum: "Efficient Network Automatic Relevance Determination"
_ICML.cc/2025/Conference — ICML 2025 poster_

### Official Review · Reviewer_YET1 · 2025-03-09

**Overall Recommendation:** 4

**Summary:**

The paper introduces **Network Automatic Relevance Determination (NARD)**, an extension of Automatic Relevance Determination (ARD) designed for linearly probabilistic models. NARD aims to simultaneously model sparse relationships between input features X and output responses Y, while capturing correlations among the outputs Y. The method employs a **matrix normal prior** with a sparsity-inducing parameter to identify and discard irrelevant features, thereby promoting sparsity in the model.

The algorithm iteratively updates both the precision matrix and the relationship between Y and refined inputs. To address computational inefficiencies associated with the high cost per iteration, the authors propose two enhancements:

1. **Sequential NARD**: Evaluates features sequentially to reduce computational overhead.

2. **Surrogate Function Method**: Uses an efficient approximation of the marginal likelihood, simplifying determinant and matrix inverse calculations.


By combining these approaches, the computational complexity is further reduced. The paper demonstrates that these methods achieve significant improvements in computational efficiency while maintaining comparable predictive performance on both synthetic and real-world datasets.


## Update after rebuttal
Thank you for your response. I will maintain my positive score.

**Claims And Evidence:**

The claims made in the paper are generally well-supported by clear evidence, as outlined below:

1. **Sparse Feature Selection via ARD Prior**
  The paper claims that placing an ARD prior on the regression coefficient matrix enables effective feature selection by identifying relevant input features for predicting outputs.

2. **Sparsity in Output Dependencies via L1 Penalty**
  The use of an L1 penalty on the precision matrix to model dependencies among outputs is a reasonable and theoretically sound approach. Sparse precision matrices are widely used in multi-output regression to capture conditional independence relationships between outputs.

3. **Computational Challenges in High Dimensions**
  The claim that standard ARD methods incur $O(d^3)$ computational costs due to matrix inversion is accurate and consistent with the literature.

4. **Efficiency of Proposed Algorithms**
  The paper introduces Sequential NARD, Surrogate NARD, and Hybrid NARD to reduce computational complexity. The stated reductions to $O(m^3+p^3)$, $O(m^3+d^2)$, and $O(m^3+p^2)$, respectively, are plausible given the described modifications (e.g., sequential updates and surrogate function approximations).

**Essential References Not Discussed:**

N/A

**Experimental Designs Or Analyses:**

The experimental designs and analyses in the paper are sound and appropriate for the stated objectives:

1. **Synthetic Data**: The use of synthetic datasets with controlled sparsity is well-suited for validating the feature selection and dependency modeling capabilities of NARD and its variants.

2. **Real-World Applications**: The use of Aging Phenotype data (evaluated via Jaccard index) and TCGA cancer data (focusing on biological associations) aligns with the goals of demonstrating NARD's applicability to high-dimensional, multi-output regression problems.

3. **Baseline Comparisons**: Comparisons with established methods like MRCE and HS-GHS are appropriate for benchmarking both computational efficiency and estimation performance. The inclusion of time as a metric strengthens the analysis of computational complexity claims.

Issues are discussed below.

**Methods And Evaluation Criteria:**

The evaluation criteria and datasets are generally appropriate for the problem. TPR and FPR on synthetic data effectively measure feature selection performance, while the Jaccard index is suitable for comparing biological associations in the absence of ground truth. The use of TCGA data validates NARD’s real-world applicability. However, the empirical evaluation has notable gaps:

1. **Limited Dataset Diversity**: The evaluation focuses heavily on biological datasets (e.g., TCGA and aging phenotype data). Including experiments on non-biological datasets (finance?) would better demonstrate the generalizability of NARD across diverse domains.

2. **Limited Baselines**: Comparisons are limited to MRCE and HS-GHS. Incorporating additional state-of-the-art baselines would strengthen validation and contextualize NARD's performance more comprehensively.


Addressing these issues would significantly enhance the robustness and completeness of the evaluation.

**Other Comments Or Suggestions:**

- Include experiments on non-biological datasets to evaluate generalizability across domains.

- Include more detailed comparisons with alternative methods beyond MRCE and HS-GHS to strengthen the empirical validation.

- Address the scalability of Hybrid NARD explicitly in larger datasets.

- Provide a more detailed analysis of how sparsity-inducing priors affect interpretability in biological applications.

- Suggestion for Images:  It would be better to move some images, such as protein networks for COAD, to the supplementary materials. This would declutter the main text and allow the remaining figures to be enlarged for improved readability.

**Other Strengths And Weaknesses:**

**Strengths:**

1. **Originality**: The paper introduces novel extensions to ARD, including Sequential NARD, Surrogate NARD, and Hybrid NARD, addressing computational inefficiencies in high-dimensional regression tasks.
2. **Significance**: The framework is highly relevant for sparse modeling in biological and genomic datasets, making it impactful for real-world applications.
3. **Clarity**: The paper clearly explains the theoretical foundations, algorithms, and experimental results, ensuring accessibility for readers familiar with Bayesian modeling.

**Weaknesses:**

1. **Numerical Stability**: Surrogate NARD exhibits instability during precision matrix estimation in high-dimensional datasets, which could limit its reliability in practice.
2. **Linear Assumptions**: The reliance on linear models may restrict applicability to scenarios involving complex non-linear relationships.
3. **Limited Dataset Diversity**: While synthetic and biological datasets are used, additional experiments on diverse real-world datasets could enhance generalizability.
4. Comparisons with baseline methods (MRCE and HS-GHS) are limited in scope. Including more state-of-the-art baselines would provide a clearer picture of NARD's relative performance.

**Questions For Authors:**

- How does Hybrid NARD balance computational efficiency and predictive accuracy compared to Sequential and Surrogate NARD? Could you provide additional insights into its practical advantages?
- Have you considered extending the framework to non-linear models or incorporating kernel methods for capturing complex relationships? If not, what challenges do you foresee?
- Could you elaborate on how the sparsity-inducing priors affect interpretability in biological applications like TCGA cancer data?
- To improve the empirical evaluation, could you expand the experiments to include non-biological datasets to assess generalizability across diverse domains, and explicitly address the scalability of Hybrid NARD on larger datasets, including runtime and performance metrics?
- Why are Hybrid NARD and Sequential NARD not included in Table 2, where the impact of data size on performance is analyzed? Including these methods would provide a more complete comparison of their scalability and efficiency relative to other approaches.

**Relation To Broader Scientific Literature:**

The paper builds on foundational concepts in Automatic Relevance Determination (ARD), introduced by MacKay (1992), extending it to multi-output regression with a matrix normal prior and sparsity-inducing penalties. It addresses computational challenges in high-dimensional settings, improving efficiency through Sequential and Surrogate methods inspired by Tipping's work (2003). The framework aligns with broader literature on Bayesian sparse modeling, such as MRCE (Rothman et al., 2010) and graphical lasso techniques (Friedman et al., 2008), while contributing novel algorithmic advancements for scalable multi-output regression.

**Theoretical Claims:**

The theoretical claims in the paper are well-supported by the proofs provided in the text. Key claims include:

1. Matrix Normal Prior and ARD Framework

2. Complexity Reductions

3. Theorem 3.1 (Sequential NARD)

4. Surrogate Function Approximation

---

> ### Author Rebuttal · Authors · 2025-04-01
>
> Thank you for your valuable feedback and constructive comments. We appreciate your positive remarks on the novelty of our methods and the clarity of the paper.
>
> >Dataset diversity
>
> We have expanded our experiments to include 2 non-biological datasets: Kaggle’s air quality dataset (https://archive.ics.uci.edu/dataset/501/beijing+multi+site+air+quality+data) and A-shares stock dataset.
>
> For the air quality dataset, we performed data imputation and timestamp alignment, then analyzed the relationships among 11 key indicators. The results show a strong correlation between PM2.5 and humidity, supporting the environmental principle that higher humidity promotes the adhesion of fine particles, leading to increased PM2.5 levels. This aligns with prior studies on atmospheric dynamics.
>
> For the A-shares dataset, we collect nearly 7 years of daily trading data and use the previous 5 days' information to predict the next day's opening price. This results in a dataset of 3032 stocks. Our experiments show that Bayesian methods, such as HS-GHS and JRNS, could not complete calculations within 4 days, while our approach demonstrate excellent scalability. Analysis of the precision matrix reveals significant block structures, indicating that stocks from the same sector or industry tend to show similar trends in price movement.
>
> Through these 2 experimental datasets, we have demonstrated the effectiveness of our method in both environmental and financial domains.
>
> Since there is no ground truth, we used MRCE as a baseline algorithm for comparison. We reported the Jaccard index as a benchmark. Additionally, we presented the computational time to highlight the computational advantages of our method.
>
> Table R4:  Associations of A-shares stocks. ($m=3032, d=60640, N=1696$)
> | Method | MRCE |CAPME |HS-GHS |JRNS | NARD |Sequential NARD | Suggorate NARD | Hybrid NARD |
> | --- | --- | --- | --- | --- | --- |--- | --- |--- |
> | # of association |97939|98335| -|-|99671|100309|99105|99475|
> | Jaccard index |-|0.869|-|-|0.881|0.891|0.893|0.890|
> | Time per iteration (second)|\~1200|\~1300| -|-|~1000|-|255|-|-|
> | Time all (h)|\~17|\~16.5|-|-|\~14.5|\~8|~5.5|\~3|
>
> >Interpretability in biological applications
>
> Sparsity-inducing priors like ARD enhance interpretability in biological applications, such as TCGA cancer data, by identifying key features. In our analysis across 7 tumor types, ARD highlighted important genes and proteins linked to signaling pathways. In Figure 3, sparsity revealed consistent pathways across cancer types, exposing cancer-specific translational effects. In Figure 4, for COAD, sparsity highlighted critical protein interactions within pathways and cross-talk between them, aiding biological interpretation. In COAD, the PI3K/AKT pathway was highlighted by the interaction between GSK3ALPHABETAPS21S9 and AKTPS473. This association indicates a key regulatory role in tumor growth and survival. The AKT signaling axis, activated by various upstream kinases like GSK3, has been implicated in colon cancer progression, making it a valuable target for further investigation and therapeutic development.
>
> >Question about Table 2
>
> We understand your concern about the exclusion of Hybrid NARD and Sequential NARD in Table 2. Table 2 shows single update step times, while Table 1 presents total computation time. Since both methods involve iterative updates with varying step times, direct comparison in Table 2 is difficult. We apologize for any confusion and will clarify this in the final version.
>
> >Linear assumptions
>
> We have considered extending the framework to non-linear models, which can be easily adapted to sparse kernel regression without significant changes to NARD. Relevant experiments and results are discussed in our rebuttal to Reviewer gDpz.
>
> >Numerical stability
>
> We appreciate your attention to this issue, which we have discussed in detail in Appendix F.4. We have provided additional analysis in our rebuttal to Reviewer gDpz, which you can refer to for a more in-depth discussion.
>
> >Discussion about Hybrid NARD
>
> We have included further analysis on this issue in our rebuttal to Reviewer AaWq, which provides a more detailed discussion.

---

### Official Review · Reviewer_gDpz · 2025-03-10

**Overall Recommendation:** 3

**Summary:**

This paper introduces the Network Automatic Relevance Determination (NARD) framework for linearly probabilistic models. It proposes three novel algorithms, i.e. Sequential NARD, Surrogate NARD, and Hybrid NARD, which significantly reduce the computational complexity. These methods maintain comparable performance on synthetic and real-world datasets, effectively handling the sparse relationships between inputs and outputs while capturing output correlations.

**Claims And Evidence:**

I didn't find any obvious problem.

**Essential References Not Discussed:**

N/A

**Experimental Designs Or Analyses:**

1. The paper only compares NARD and its variants with two baseline methods (MRCE and HS-GHS). Given the large number of methods available for sparse multivariate regression and graphical model estimation, this limited comparison may not fully establish the superiority of the proposed methods.
2. The authors assume linear relationships in their models. While this is a common starting point, real-world data, especially in biology, may contain complex nonlinear relationships. The experimental designs do not explore how well the methods perform in the presence of nonlinearity.
3. The Surrogate NARD method shows instability in computations, especially when dealing with large, high-dimensional real-world datasets.

**Methods And Evaluation Criteria:**

Generally speaking, the proposed methods and evaluation criteria in the paper make sense for the problem at hand. Some of my concerns might stem from whether the proposed methods are SOTA. Given that the benchmark methods for comparison are relatively limited, as far as I know, there should be numerous methods for handling sparse multivariate regression and learning sparse graph structures. It would be better if the authors could explain why they chose MRCE and HS - GHS as the comparison techniques.

**Other Comments Or Suggestions:**

N/A

**Other Strengths And Weaknesses:**

N/A

**Questions For Authors:**

N/A

**Relation To Broader Scientific Literature:**

In the related work section, the authors establish connections between the content of this paper and the broader scientific literature. Additionally, in the experimental part, authors compares NARD and its variants with two baseline methods (MRCE and HS-GHS).

**Theoretical Claims:**

Sorry, I didn't check the correctness of proofs.

---

> ### Author Rebuttal · Authors · 2025-04-01
>
> Thank you for your thoughtful feedback. We appreciate your positive remarks and the concerns raised, which will help us refine both the theoretical and empirical aspects of our work.
>
> >Baseline comparison
>
> As suggested, we add additional baseline methods: CAPME[1] and JRNS[2]. MRCE and CAPME are frequency-based representative methods, while HS-GHS and JRNS are Bayesian sampling-based algorithms. Our experiments show that NARD still outperforms these methods, which further strengthens the case for the proposed approach. The results are shown in Table R2.
>
> In the original paper, we carefully selected MRCE and HS-GHS as the comparison methods because they represent two well-established approaches in the field: MRCE is a frequentist method, and HS-GHS is a Bayesian approach. Both of these methods have been compared with several other techniques in their respective papers, and in those comparisons, MRCE and HS-GHS consistently performed well. We believe these methods serve as strong baselines for our study, providing a comprehensive comparison between frequentist and Bayesian paradigms in sparse multivariate regression and graphical model estimation.
>
> Table R2: Performance Comparison of Various Methods.
> | Method | d | m | N | TPR | FPR | Time |
> | --- | --- | --- | --- | --- | --- |--- |
>  |MRCE | 5000| 1500|1500 | 0.9083| 0.0072 | 53 |
> |**CAPME** | 5000| 1500|1500 | 0.8972| 0.0124 | 52 |
> |HS-GHS | 5000| 1500|1500| 0.9463| 0.0033 | >3000 |
> |**JRNS** | 5000| 1500|1500| 0.9485| 0.0037 | >3000 |
> |NARD | 5000| 1500|1500 | 0.9483| 0.0062 | 49|
> |Sequential NARD | 5000| 1500| 1500| 0.9459| 0.0067 | 35|
> |Surrogate NARD |5000 |1500 |1500  | 0.9462| 0.0072 | 31|
>  |Hybrid NARD | 5000| 1500 | 1500| 0.9471| 0.0068 | 23|
>
> We also include the results of experiments on aging phenotype data as shown in Table R3.
>
> Table R3:  Associations under different algorithms.
> | Method | MRCE |CAPME |HS-GHS |JRNS | NARD |NARD(Polynomial) | NARD(RBF) |
> | --- | --- | --- | --- | --- | --- |--- | --- |
> | # of association | 15330 |15094| 14983|15066| 15101|15094| 15072|
> | Jaccard index | 0.979 |0.977|-| 0.988 |0.988 | 0.990 |0.989 |
>
> [1] Covariate-adjusted precision matrix estimation with an application in genetical genomics, Biometrika 2013.
>
> [2] A generalized likelihood based Bayesian approach for scalable joint regression and covariance selection in high dimensions, Statistics and computing 2022.
>
> >Non-linearity
>
> In our paper, Section 6.1 focuses on synthetic data, which follows a linear structure by design. However, in Sections 6.2 and 6.3, we evaluate our method on real-world biological datasets with more complex, nonlinear relationships. Despite these nonlinearities, our approach performs well, suggesting that our sparse linear approximation effectively captures dominant interaction patterns. We will clarify this fact more clearly in the final version.
>
> Additionally, our method can be naturally extended to address nonlinearity through kernel method. Specifically, we consider the model
> $$
> Y = W\Phi(X) + \mathcal{E}, \quad \Phi(\cdot) \in { \text{Polynomial, RBF, ...} }
> $$
> where $\Phi(X)$ represents a nonlinear feature mapping that transforms the input space into a higher-dimensional space, allowing for more flexible modeling of complex relationships.
>
> To explore this extension, we consider 2 different kernel functions: the polynomial kernel and the Gaussian (RBF) kernel. We evaluate the performance on a real-world aging phenotype data.
>
> As shown in Table R3, our approach with the polynomial and RBF kernels demonstrates competitive performance, achieving high Jaccard index values. The results are consistent with our expectation that kernel-based extensions allow the model to capture more complex, nonlinear relationships in the data, further validating the robustness and flexibility of our method.
>
> >Numerical stability in Surrogate NARD
>
> We appreciate your attention to this issue, which we discuss briefly in Appendix F.4. It arises from numerical challenges encountered during the iterative optimization process, particularly when estimating the precision matrix. For large datasets, the covariance matrix can be ill-conditioned, leading to instability in the precision matrix estimation. This stems from the inherent properties of the data. As we mentioned in the discussion, one potential solution is to apply a more robust initialization for the precision matrix, which may help mitigate this issue. We will explore this aspect more deeply in future work.

---

### Official Review · Reviewer_AaWq · 2025-03-11

**Overall Recommendation:** 3

**Summary:**

This paper introduces Network Automatic Relevance Determination (NARD), an extension of Automatic Relevance Determination (ARD) designed for multiple-output regression in high-dimensional settings. NARD integrates a matrix normal prior with a sparsity-inducing mechanism to simultaneously select relevant input features and capture output dependencies.

**Claims And Evidence:**

Yes.

**Essential References Not Discussed:**

I think all related works have already been cited.

**Experimental Designs Or Analyses:**

I have checked the soundness and validity of the experimental designs and analyses, and I find them reasonable. For example, in the synthetic data experiments, the covariance and precision matrices were generated using an Erdős-Rényi random graph, ensuring a structured yet realistic sparsity pattern. The metrics used (TPR and FPR) are appropriate for evaluating feature selection.

**Methods And Evaluation Criteria:**

Yes.

**Other Comments Or Suggestions:**

Theorem 3.1 could benefit from additional explanation for edge cases and it would be helpful to provide more details on how hyperparameters were tuned.

**Other Strengths And Weaknesses:**

The paper presents a novel extension of Automatic Relevance Determination (ARD) by incorporating a matrix normal prior. The introduction of Sequential NARD and Surrogate NARD significantly improves computational efficiency over traditional ARD methods, which is a notable strength.

The paper could provide more clarity on hyperparameter selection. Additionally, the paper could further discuss potential trade-offs in Hybrid NARD.

**Questions For Authors:**

The Hybrid NARD approach combines Sequential and Surrogate NARD. Are there any cases where this hybrid method performs worse than using either method individually? How does it balance efficiency vs. accuracy in practice? Do you have more experiments for this point?

**Relation To Broader Scientific Literature:**

The paper extends Automatic Relevance Determination (ARD) by incorporating a matrix normal prior to model both feature sparsity and output dependencies, addressing limitations in traditional ARD. NARD in this paper further improves computational efficiency by introducing Sequential NARD, which employs a greedy approach that sequentially adds and removes features, and Surrogate NARD, which introduces a surrogate function to approximate the marginal likelihood.

**Theoretical Claims:**

I have checked the theoretical derivations in Sections 3 and 4, and I find them reasonable. For example, in Section 4, I verified Lemma 4.2, which establishes an upper bound on $ \text{Tr}[g(W)] $ using a majorization argument. The proof correctly applies Lipschitz continuity, with $ L = 2\|XX^\top\| = 2\rho $, and a first-order Taylor approximation to derive the bound. The use of $ \rho $ (the largest eigenvalue of $ XX^\top $) ensures a valid approximation, confirming the correctness of the inequality.

---

> ### Author Rebuttal · Authors · 2025-04-01
>
> Thank you for your thoughtful and constructive feedback on our submission.
> >Edge case of Theorem 3.1
>
> Recall the Theorem 3.1, $s_i$ is called the sparsity and $q_i$ is known as the quality of $\varphi_i$, The sparsity measures the extent to which basis function overlaps with the other basis vectors in the model, and the quality represents a measure of the alignment of the basis vector with the error between the training set values and the vector of predictions that would result from the model with the vector excluded. The term $\eta_i = \text{Tr}(q_i q_i^{\top}V^{-1}) - m s_i$ actually measures the trade-off between the alignment quality of the basis vector and its sparsity in relation to the covariance structure. For $L(\alpha_i)$, when $\eta_i > 0$, the function exhibits an initial increase followed by a decrease, with the maximum value occurring at a stationary point. When $\eta_i \le 0$, the process is monotonically increasing, and the maximum value is asymptotically approached as $\alpha_i \rightarrow \infty$, consistent with the proof of Theorem 3.1. Furthermore, as $\alpha_i \to \infty$, the part of $L(\alpha_i)$ that depends on $\alpha_i$ diminishes, and $L(\alpha_i) = 0$ represents the situation where the corresponding feature can be pruned from the model.
>
> >Tuning the $\lambda_{\text{glasso}}$ Parameter in the ARD Framework
>
> To select the optimal $\lambda$,  we employ a 5-fold cross-validation procedure. The dataset is partitioned into 5 disjoint subsets, and in each iteration, 1 subset is held out as the validation set while the remaining 4 subsets are used for model estimation. The objective function for selecting $\lambda_{\text{glasso}}$ is defined as:
> $$
> \lambda_{\text{glasso}} = \arg \min_{\lambda} \sum_{l=1}^5
> \bigg[
>     \textbf{Tr}(\tilde{V_l}\Omega_{-l}) -\log |\Omega_{-l}| + \lambda \sum_{\substack{i \neq j}} |\omega_{ij}|
> \bigg].
> $$
> Here $\tilde{V_l}$ is the empirical covariance estimator computed from the training data excluding the $l$-th fold, and $\Omega_{-l}$ is the estimated precision matrix based on this subset.
> The log-likelihood is computed for each fold, and the $\lambda$ that maximizes the cross-validated log-likelihood is chosen. A grid search is performed over a range of candidate values for $\lambda$ , and the value that yields the best performance across all folds is selected for the final model and evaluation.
>
> >Discussion about Hybrid NARD
>
> In our experiments, we did not observe cases where the Hybrid NARD underperformed compared to either Sequential or Surrogate NARD individually. Despite incorporating approximations, the hybrid approach consistently provided a good balance between accuracy and efficiency. In the synthetic data shown in Table 1, we found that NARD and its variants performed similarly, with no significant differences. However, in terms of time efficiency, the Hybrid NARD was approximately twice as fast as the standard NARD. This aligns with our theoretical expectations, where combining sequential optimization with surrogate modeling helps leverage their respective advantages.
>
> We acknowledge that performance trade-offs may become more pronounced in extreme cases. For smaller datasets, we hypothesize that the hybrid approach may not always outperform NARD. For instance, when tested with $d=80, m=50,N=100$ and $d=50, m=20,N=50$, the results (shown in Table R1) demonstrated that Hybrid NARD still performed well, highlighting the robustness of our method. Despite our efforts to identify challenging edge cases, the algorithm exhibited a notable degree of stability.
>
> We appreciate your suggestion and will explore these scenarios further in future work.
>
> Table R1: Performance Comparison of Our Methods on Small datasets.
> | Method | d | m | N | TPR | FPR |
> | --- | --- | --- | --- | --- | --- |
>  | NARD | 80| 50| 100 | 0.9695| 0.0031 |
>  | Sequential NARD | 80| 50| 100| 0.9689| 0.0033 |
> |Surrogate NARD |80 |50 |100| 0.9693| 0.0029 |
>  |Hybrid NARD | 80| 50 | 100| 0.9689| 0.0035 |
>  | NARD | 50| 20|50 | 0.9542| 0.0051 |
>  | Sequential NARD | 50| 20| 50| 0.9544| 0.0046 |
> |Surrogate NARD |50 |20 |50| 0.9540| 0.0050 |
>  |Hybrid NARD | 50| 20 | 50| 0.9540| 0.0049 |

---

> > ### Comment · Reviewer_AaWq · 2025-04-03
> >
> > Dear the Authors,
> >
> > Thank you for your thorough response, which addresses most of my concerns. Therefore, I decided to increase the rating accordingly. I encourage the authors to revise the manuscript as discussed above.

---

> > > ### Author Response · Authors · 2025-04-03
> > >
> > > Thank you for your feedback and for increasing the rating. We appreciate your suggestions and will revise the manuscript accordingly. Your comments have been very helpful in improving our work.

---

### Decision · Program_Chairs · 2025-05-01

**Decision:**

Accept (poster)

**Comment:**

The paper re-visits classical probabilistic linear models (multi-input, multi-output) which can naturally identify relevance of model parameters. Specifically, a method is devised that promotes sparsity in the linear coefficients while capturing correlations among output variables.

The reviewers were consistently positive about the paper and recommended to accept it. The method seems sound and compares favorably to existing baselines.

The reviewers raised some concerns which have been discussed in the rebuttal phase and which shall be addressed by the authors in the revision of the paper.